# The role of learned song in the evolution and speciation of Eastern and Spotted towhees

Ximena León Du'Mottuchi[1,2], Nicole Creanza[1,2]*

1 Department of Biological Sciences, Vanderbilt University, Nashville, Tennessee, United States of America, 2 Evolutionary Studies, Vanderbilt University, Nashville, Tennessee, United States of America

* nicole.creanza@vanderbilt.edu

## Abstract

Oscine songbirds learn vocalizations that function in mate attraction and territory defense; sexual selection pressures on these learned songs could thus accelerate speciation. The Eastern and Spotted towhees are recently diverged sister species that now have partially overlapping ranges with evidence of some hybridization. Widespread community-science recordings of these species, including songs within their zone of overlap and from potential hybrids, enable us to investigate whether song differentiation might facilitate their reproductive isolation. Here, we quantify 16 song features to analyze geographic variation in Spotted and Eastern towhee songs and assess species-level differences. We then use several machine learning models to measure how accurately their songs can be classified by species. While no single song feature reliably distinguishes the two species, machine learning models classified songs with relatively high accuracy (random forest: 89.5%, deep learning: 90%, gradient boosting machine: 88%, convolutional neural network: 88%); interestingly, species classification was less accurate in their zone of overlap. Finally, our analysis of the limited publicly available genetic data from each species supports the hypothesis that the species are reproductively isolated. Together, our results suggest that small variations in multiple features may contribute to these sister species' ability to recognize their species-specific songs.

## Author summary

Songbirds learn their songs through imitation; these songs are important in mate selection and thus could affect evolution. Multiple factors, including genetics, cultural transmission, and environmental conditions, could influence the song differences between individuals, and some song variations might be more attractive than others. These variations in song can accumulate over time, leading to the emergence of distinct dialects and regional variations, which can potentially contribute to reproductive isolation and speciation. Thus, in order to shed light

**Data availability statement:** All analysis code and data are available at https://github.com/CreanzaLab/TowheeAnalysis, including metadata about each song recording.

**Funding:** X.L.D. and N.C. were supported by the National Science Foundation Division Of Integrative Organismal Systems (IOS-2327982). X.L.D. was supported by a Graduate Research Fellowship from the National Science Foundation (nsf.gov; DGE-2139839). The funders did not play any role in the study design, data collection and analysis, decision to publish, or preparation of the manuscript.

**Competing interests:** The authors have declared that no competing interests exist.

onto the incredible diversity of songbird species, we aim to uncover evolutionary patterns in birdsong to improve our understanding of its role in speciation. To this end, we used statistical analyses to investigate the patterns of song variation that exists in the Spotted and Eastern towhees, a sister-species pair whose ranges overlap, and we use various machine learning algorithms trained on 16 song features to assess the degree of song distinguishability between the two species. Our results show that no single song feature reliably distinguishes the two species; however, combinations of these features could classify songs with high accuracy. These findings suggest that song could play some role in the ability of these birds to recognize their own species' songs.

## Introduction

In the oscine songbirds, an individual's song is a set of learned vocalizations used primarily for attracting mates and defending territories, and the characteristics of these songs vary widely across different species [1]. Song also varies within a species, with some differences in birdsong characteristics occurring at large spatial scales [2]. This long-range geographic variation could be influenced by a combination of genetic and environmental factors, such as adaptation to local habitats and ecological conditions [3]. Furthermore, under the assumption of isolation-by-distance due to spatially limited dispersal, songs are expected to accumulate differences gradually over time and become increasingly different with geographic distance [4]. In vocal learners, such as songbirds, geographic variation can also accumulate through cultural transmission of learned vocalizations because of many factors, including, but not limited to, physical isolation and divergence in sexual selection [3].

Divergence in traits that are under sexual selection is important for premating reproductive isolation as two species diverge [5], since mate preference based on these traits can limit interpopulation mating, which is beneficial if hybrid offspring have lower fitness [6–10]. For example, in a study of chorus frogs (*Pseudacris nigrita nigrita* and *Pseudacris triseriata feriarum*), the (unlearned) calls of the two species show significant differences in pulse rate and pulse number in sympatry, suggesting that these characteristics may have diverged via reproductive character displacement [11]. A similar pattern has been observed for a learned behavior: Ratcliffe and Grant [12] found that although songs of *Geospiza fortis* and *Geospiza fulginosa* had very similar syllable structure and timing, individuals still preferred their own species' songs in a playback experiment, suggesting that they are likely using subtle song differences to inform their mate choice. Since song functions in species recognition, these song differences can inhibit birds from choosing mates from different locations with less familiar song characteristics or from other subspecies [5,10,13]. In theory, a rapid accumulation of learned song changes can lead to reproductive isolation, accelerating speciation [9].

Hybridization can occur when two genetically distinct populations come into contact with one another and mate [14,15]. In some hybrid zones, the hybrids between

two species have high fitness and are not selected against, so the hybrid zone can be quite large [5,16]. However, cross-mating does not necessarily lead to the populations merging into one. For example, selection that favors the prevalent phenotype of a given area constrains the width of the hybrid zone because individuals that disperse in the range of the other species could exhibit inferior fitness [5,17]. A study that used playback experiments on *Streptopelia vinacea* and *Streptopelia capicola*—a pair of non-vocal-learning dove species that hybridize—found that males in allopatric populations responded more to conspecific vocalizations than heterospecific or hybrid vocalizations whereas males in the hybrid zone responded equally to all three vocalizations, suggesting that non-hybrid individuals may have reduced fitness in areas that the other species occupies and that hybrids may have reduced fitness in areas outside the hybrid zone [18]. Further, separation of species can be maintained by reinforcement if the strength of selection is greater for the characteristics that fall in the extremes rather than those of the hybrids [5], such that premating isolation is strengthened. In other words, if sexual selection favors more extreme characteristics, then hybrids will have lower mating success. For example, a study of the same two species of chorus frogs (*P. nigrita* and *P. feriarum*) showed evidence that hybrid fitness was significantly reduced, suggesting that reinforcement, via sexual selection against hybrid males, is the primary mechanism behind the reproductive character displacement in these species [19].

Further, searching for a mate can be costly to females if conspecific males are difficult to find in a given area. Therefore, in a zone of overlap between two closely related species, females of the rarer species could have an increased tendency to hybridize with mates of the more common species, particularly if the signal of the heterospecific male is similar to that of the conspecific male [5,20,21]. Stability of a narrow hybrid zone can be maintained via spatial segregation of the two populations: as the hybrid zone is traversed, individuals of one species will become less common, leading them to mate more frequently with the other species and produce hybrids with reduced fitness [5,22].

Cultural transmission of songs can also lead to individuals producing heterospecific songs or mixed songs at the contact zone through heterospecific song copying, which can then lead to hybridization and introgression [23]. A study of the pied flycatcher (*Ficedula hypoleuca*) and collared flycatcher (*Ficedula albicollis*), both vocal learners, found that in areas of sympatry the pied flycatchers used mixed songs that resembled songs of the collared flycatcher, suggesting that pied flycatchers were using heterospecific copying that minimized species differences in sympatry [24]. A study of two *Setophaga* warblers also observed that the species' song features were more similar in the hybrid zone [25]. Since learned traits can be transmitted independent of genotype, cultural traits can either foster or limit assortative mating [26]; thus, cultural transmission of song has the potential to lead to both reproductive isolation or hybridization, making it a complex mechanism of evolution.

The Spotted towhee (*Pipilo maculatus*) and the Eastern towhee (*Pipilo erythrophthalmus*) are sister species of the Passerellidae family of oscine songbirds. It is estimated that the Eastern and Spotted towhees diverged 280,000 years ago [27,28]. Evidence from the geological history of the Great Plains, in combination with patterns of the current geographic distribution of the towhees and their character gradients, suggest that present hybridization of these sister species in the Great Plains is a result of secondary contact that occurred after the Pleistocene glaciation fragmented their breeding range and accelerated speciation [5,27,29]. They were previously considered one species, the Rufous-sided towhee, and the idea that the Spotted and Eastern towhees should be considered separate species has been contentious for many years. These sister species were separated into their respective groups primarily due to distinctions in their geographic distribution, genetic data, and morphological characteristics such as plumage, sexual dichromatism, and eye color [30]. Secondarily, early studies noted some differences between songs in the eastern and western populations [29].

The breeding range of the Spotted towhee lies on the western side of the United States extending into Mexico and southern Canada, while the breeding range of the Eastern towhee lies on the eastern side of the United States and southern Canada. Hybridization of the Spotted and Eastern towhees has been reported in the Great Plains area, particularly in Nebraska, where towhees had been noted as exhibiting 'intermediate' songs [29]. Further, Sibley and West [29] used

plumage as an index for hybridization by scoring birds in the Great Plains based on spotting of the wing coverts (males) and color of the head and back (females). Although hybridization occurs, Spotted and Eastern towhees are still visually distinguishable as separate species and remain genetically dissimilar. While these species share many similarities in their vocalizations, subtle but consistent differences in their songs could allow individuals to use auditory information alongside plumage features to distinguish between the different species and identify potential conspecific mates, allowing separation of species to be maintained, a form of behavioral premating isolation.

The typical song of both the Spotted towhee and Eastern towhee is a loud and clear "drink-your-tea", composed of short introductory notes and a fast trill (Fig 1). Previous studies suggest that the Eastern towhees tend to have songs that are more variable and complex, with a greater number of syllables, whereas the Spotted towhees tend to have fewer syllables or no introductory syllables with a faster trill [31,32]. In this study, we aim to distinguish any geographic patterns in the song characteristics of the Spotted towhee and Eastern towhee, which have not been quantified between the

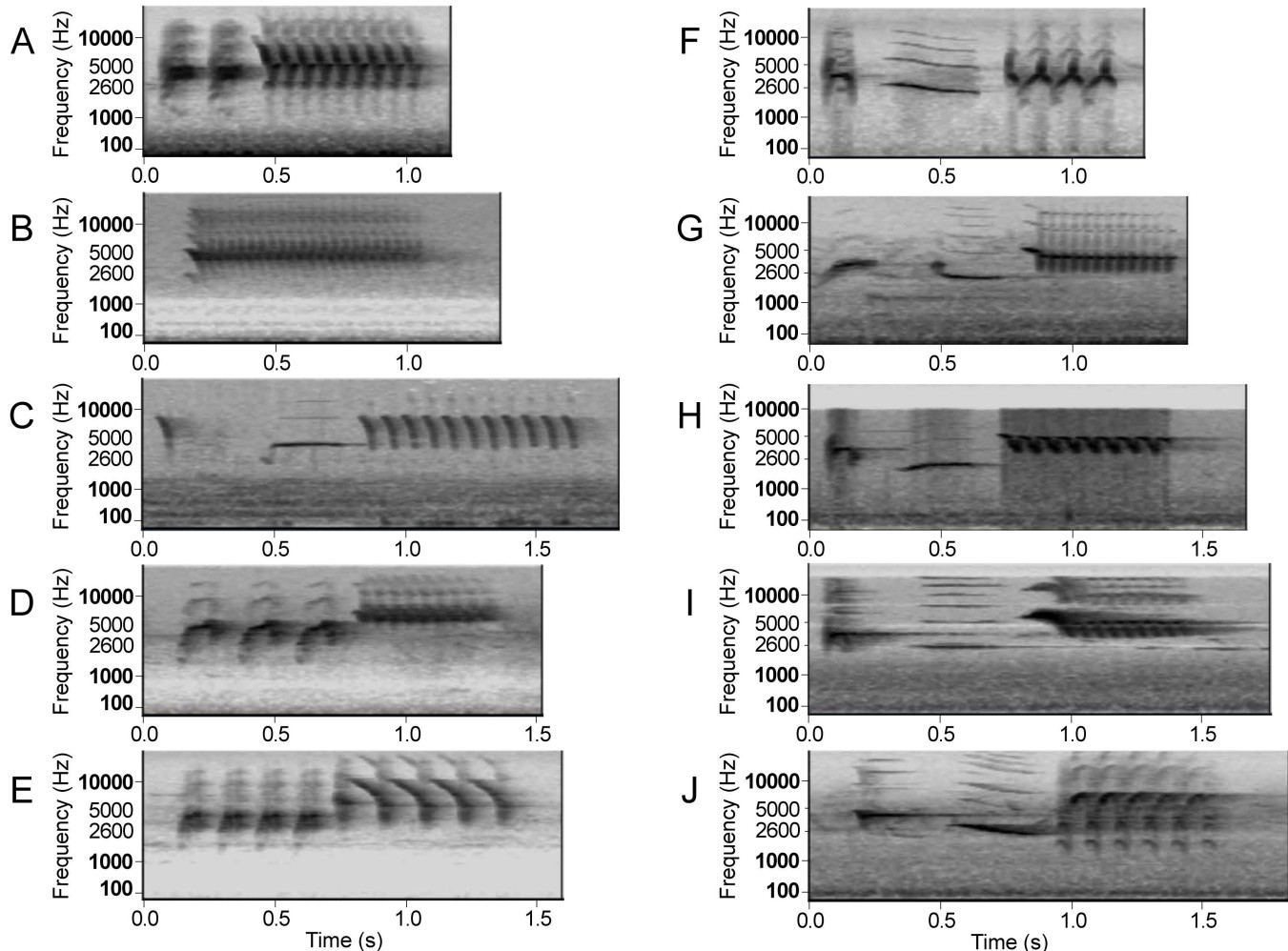

**Fig 1. Example spectrograms of Spotted and Eastern towhee songs.** Examples of Spotted towhee songs (**A-E**) and Eastern towhee songs (**F-J**) from Macaulay Library (ML) and Xeno-canto (XC). The repeated syllable at the end of each song is often called the 'trill'. Recording ID numbers: (**A**) ML191126; (**B**) ML90009051; (**C**) XC575461; (**D**) XC577591; (**E**) XC577593; (**F**) ML15276; (**G**) ML200973; (**H**) ML54283841; (**I**) ML76939751; (**J**) ML450438861; see metadata (https://github.com/CreanzaLab/TowheeAnalysis) for detailed recording information.

two species. By analyzing the songs of each species individually as well as together as a two-population cline, we aim to understand if there are song patterns that are continuous across the geographic ranges of the Eastern and Spotted towhees or if there are discontinuities in the geographic distribution that clearly distinguish the songs of the two species from one another. Additionally, we use machine learning to investigate whether the two species differ in their song characteristics such that birds could potentially compare multiple song features to distinguish between conspecific and heterospecific individuals.

## Methods

### Generating a database of song bouts

We downloaded recordings of Spotted and Eastern towhees from the Macaulay Library (Cornell Lab of Ornithology 2009) by requesting access to the relevant song files, as well as from Xeno-canto [33] using the WarbleR package in R to query the database [34]. We filtered the recordings to eliminate those without song by specifying "Sounds/Type = Song". We also requested recordings marked as "Spotted × Eastern Towhee (hybrid)" and "Spotted/Eastern Towhee (Rufous-sided Towhee)" from the Macaulay Library to analyze separately as a "hybrid/unsure" category—of the 27 hybrid/unsure recordings we obtained, 24 recordings had photos and/or a description of the morphology of the individual (Table A in S1 Text). For further analysis, we constructed a spreadsheet of metadata for each recording, which included species, date, location, latitude, longitude, and recordist. To minimize the chance of including repeated song recordings of the same bird, we discarded any duplicate entries with the same recordist, date, and location. For recordings that named a location but not geographic coordinates, we estimated the latitude and longitude by identifying the location on a map. We opened each recording file in Audacity version 3.1.3 (https://www.audacityteam.org/) to manually extract one song bout from each recording. Files from Xeno-canto are generally stored in mp3 format and files from Macaulay Library in wav format; we resampled recordings at 44,100 Hz if they were recorded at a different sampling rate and exported each bout as a.wav file to standardize the recordings for analysis, and we noted whether the sampling rate and/or file format had been converted. During resampling, Audacity implements a low-pass filter at half of the sampling rate to prevent aliasing. We considered the breeding season to be between April and August based on the "Breeding" section of the Birds of the World database [35,36], and we conducted analyses both on recordings from any month (see Results) and on recordings from only the breeding season months (see S1 Text).

To assign the zone of song overlap between the ranges of the two species, for each degree longitude we plotted the proportions of Eastern and Spotted towhees relative to the total number of both species combined, which allowed us to identify where the two species co-occur most frequently and how their relative abundances shift geographically. In other words, at each degree longitude we calculated the number of recordings of Spotted towhees divided by the total number of breeding-season recordings of both species combined, and we repeated this calculation at each degree longitude for the Eastern towhees. We then plotted these proportions on a graph where the x-axis represents longitude and the y-axis represents the proportion of each species (Fig A in S1 Text). At the most eastern and western longitudes, the recordings were all from Eastern and Spotted towhees, respectively. Based on our findings, we chose to identify the area between 102°W and 91°W as the zone of song overlap, which contained the majority of the area where recordings of both species seemed to co-occur (Fig A in S1 Text). In addition, we downloaded from eBird all sightings of putative hybrids (eBird query: "Spotted × Eastern Towhee (hybrid) - Pipilo maculatus × erythrophthalmus") during the breeding season and plotted their locations; we observed that reported hybridization events primarily occurred in this region (Fig B in S1 Text). We also determined an approximate frequency of hybridization in the zone of overlap by finding the area with the highest density of hybrid sightings between 2013 and 2023 (Nebraska) and dividing the estimated number of unique sightings of hybrid towhees in Nebraska by the number of unique Spotted towhee or Eastern towhee sightings in Nebraska during the breeding season, filtering out those from the same date, latitude, and longitude.

## Song analysis (syllable features and syllable segmentation)

To begin analyzing the features of the syllables within these bouts, we used the Chipper software [37], which was designed to facilitate syllable segmentation and analysis of field recordings with different levels of background noise. This software performs automated syllable segmentation by attempting to parse each song into syllables, defined as periods of sound separated by periods of silence, and then allowing the user to adjust the signal-to-noise ratio, enable high-pass and low-pass filters, and modify the segmentation when syllables have been incorrectly parsed, such as when background noise occurs during the inter-syllable silences (Fig 2). To prevent lower-amplitude syllables from being segmented incorrectly, we normalized the amplitude across each song. Additionally, we adjusted the signal-to-noise threshold, minimum syllable duration, and minimum silence duration to most accurately define the syllables in the song. Since our recordings are from nature and have wide variation in their background noise profiles, we view each spectrogram to assess what is background noise versus the bird singing and manually adjust these parameters so that the syllables are parsed in a way that appears to be most accurately defined with minimal noise; this procedure has been shown to have high within-user repeatability and high between-user reproducibility [37]. If any syllables appear to be parsed incorrectly after this procedure, Chipper allows the user to manually modify the syllable segmentation. If, when visualized in Chipper, we determined that the song overlapped with other birds singing or had excessive background noise, we removed the recording from the analysis. We provide our code for statistical analysis and visualization of the song feature data and metadata about each song recording at https://github.com/CreanzaLab/TowheeAnalysis.

After segmentation, we examined a subset of 20 song bouts per species in Chipper to estimate a noise threshold, or the minimum number of matrix elements in the spectrogram that a note must contain in order to not be discarded as noise (Spotted towhee = 87, Eastern towhee = 70, hybrid/unsure = 65), and syllable similarity threshold, or the percent syllable overlap that determines whether two syllables are considered the same or different (Spotted towhee = 26.6, Eastern towhee = 25.5, hybrid/unsure = 25.0), using Chipper's "Noise Threshold" and "Syllable Similarity Threshold" widgets, respectively. We calculated these values for both species to allow Chipper to make more accurate measurements for syllable features and song syntax specific to our set of Spotted and Eastern towhee songs. We then ran the song analysis function in Chipper, which uses the spectrogram and the syllable segmentation data to provide automated measurements of numerous song features, which were used for statistical analysis. We used 16 song-feature outputs from Chipper: bout duration (ms), number of syllables, rate of syllable production (calculated as the number of syllables divided by bout duration (1/ms)), smallest syllable duration (ms), largest syllable duration (ms), average syllable duration (ms), number of unique syllables, degree of syllable repetition (calculated as the number of syllables per number of unique syllables), average syllable upper frequency (Hz), average syllable lower frequency (Hz), maximum syllable frequency (Hz), minimum syllable frequency (Hz), overall syllable frequency range (Hz) (calculated as the overall maximum syllable frequency minus the overall minimum syllable frequency), largest syllable frequency range (Hz), smallest syllable frequency range

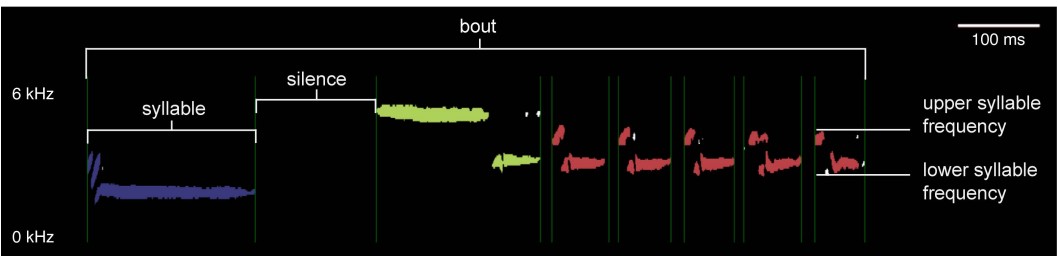

**Fig 2. Spectrogram illustrating the definition of specific song elements.** On this spectrogram of an Eastern towhee song, we indicate a syllable, silence, bout, maximum syllable frequency, and minimum syllable frequency. The color of the syllable indicates its unique syllable type. Spectrogram generated from Macaulay Library recording 52506381, https://macaulaylibrary.org/asset/52506381.

(Hz), and average syllable frequency range (Hz) (calculated as the average of the maximum minus the minimum frequency of each syllable); see [37] and the Chipper manual for more detail on song feature calculations. We then analyzed these song features across 1067 Spotted towhee song bouts, 1718 Eastern towhee song bouts, and 27 "hybrid/unsure" song bouts, using various statistical methods to assess geographic patterns and variation in songs and machine learning approaches to investigate the degree to which songs can be accurately classified using song feature data. We use multiple different machine learning methods to assess whether there is convergence in the results of the models or whether there are potential idiosyncrasies tied to each unique method.

### Data Preparation

To investigate the geographic variation in birdsong of the Spotted and Eastern towhees, we first determined that the song features were not normally distributed, so we calculated the natural log of each feature for analysis. Our analysis thus contained a total of 1067 Spotted towhee samples and 1718 Eastern towhee samples and data for 16 song features. To visualize the data, we plotted each log-transformed feature against the longitude and latitude coordinates of its recording (Figs C–E in S1 Text). All the base maps used in the figures were created using Natural Earth data (http://www.naturalearthdata.com/about/terms-of-use/) via the rnaturalearth package in R.

### Generalized Linear Models

To assess the relationship between the song feature data and potential interacting variables, we used generalized linear models (GLM). First, we generated a dataframe that includes the song feature data, geographic coordinates, and species classification for all the samples. We also include the overall conversion status of the recording files for each sample, which was denoted as 'unconverted' if the file was not converted before analysis and 'converted' if the original recording file had to be converted to a.wav file (generally from.mp3 or.m4a) or if the sampling rate had to be converted to 44,100 Hz. We then use the 'glm' function (R package: stats) to fit GLMs to our raw song feature data, using a log link function and including scaled longitude, scaled latitude, species, and the overall file conversion status as fixed effects.

### Linear Discriminant Analysis of song feature data

Using the raw song feature data, we conducted a Linear Discriminant Analysis (LDA) that uses the 16 song features to maximize the separability of the two species and minimize the variation within each species class using the 'lda' function from the MASS package [38] in R. We obtained a random subsample (75% for training and 25% for testing) of the entire data set and then downsampled the training set to obtain a balanced subsample of Eastern towhees and Spotted towhee song bouts ($N_{Spotted\_towhee} = 796$ and $N_{Eastern\_towhee} = 796$). Thus, the model was trained on 1592 total song bouts and tested on a subset of 697 song bouts that were not in the training set. We calculated the overall model accuracy $\left( \frac{number\ of\ correct\ classifications}{total\ number\ of\ songs\ tested} \right)$, balanced accuracy $\frac{1}{2} \left( \frac{number\ of\ correct\ Spotted\ towhee\ classifications}{total\ number\ of\ Spotted\ towhee\ songs\ tested} + \frac{number\ of\ correct\ Eastern\ towhee\ classifications}{total\ number\ of\ Eastern\ towhee\ songs\ tested} \right)$, and Cohen's kappa ($\kappa$) using the 'confusionMatrix' function from the caret package [39] in R for this model and all other classifier algorithms in this manuscript. We also used a permutation test to assess whether each machine learning model performed better than random chance. To do this, we randomly shuffled the species labels and compared the permuted accuracy of 1000 permutations to the actual accuracy of the model.

For these analyses and all other classifiers described below, we also assessed the effect of file conversion status on the results (see Supplemental Methods in S1 Text).

### Principal components and Procrustes analyses of song feature data

We used a Principal Component Analysis (PCA) to reduce the dimensionality of the dataset and visualize the variation in the song data, and we measured the proportion of variance explained by each principal component (PC). Using the

'prcomp' function in the stats package in R, we centered and scaled the log-transformed data and performed the PCA. We assessed whether the two species could be distinguished in PC space by testing a simple linear classifier using the 'lda' function in the MASS package in R. This type of Discriminant Function Analysis finds a linear function that best discriminates the 'Species' classification of the song recordings based on PC1 and PC2. In other words, this classifier finds a line through the two-dimensional PC-space such that songs on one side of the line are more likely to be from Spotted towhees and on the other side of the line are more likely to be from Eastern towhees, indicating the separability of the species after PCA [40]. Further, to determine whether the PCA showed geographic structuring, we followed up with a Procrustes analysis using PC1 and PC2 from the PCA of our song data. For this analysis, the matrix of PC scores was rotated and transformed onto the target matrix containing the longitude-latitude coordinates of the samples, such that the sum of squared distances between the corresponding points of the transformed and target matrix was minimized and plotted on a map using the 'procrustes' function from the vegan package [41] in R.

### UMAP visualization

We used a Uniform Manifold Approximation and Projection (UMAP) as another method of reducing dimensionality and visualizing variation in our song-feature data. We rescaled the log-transformed data using the same technique as in the PCA, and we used the 'umap' function in the umap package [42] in R to generate a two-dimensional projection of the data. Since most of the points occupy a single cluster in the resulting UMAP projection, we then tested a simple linear classifier as above to determine how well the songs could be separated by species in the UMAP projection. We included the sample of 27 "hybrid/unsure" towhees in the UMAP analysis, but we excluded these points when we tested the species classifier.

### Random forest model

We used a random forest model (RFM) to assess how accurately Spotted and Eastern towhees can be distinguished from one another by their song features. First, we log-transformed the song feature data. Next, we separated the recordings into a training set and a test set in multiple ways, as explained below, and trained the random forest classifier on the 16 song features from the training set containing species identifiers, using the ranger package [43] in R. For each model, the algorithm then averaged multiple decision trees to make predictions of the species identity in the test set. We used our random forest algorithm to train two models: 1) a model trained on a geographically unbiased subset of the entire data set and 2) a model trained on a subset of song bouts obtained from the non-overlap zone. For the former model, we obtained a random subsample (75% for training and 25% for testing) of the entire data set, and we then downsampled the training set to obtain a balanced subsample of Eastern towhees equal to the number of Spotted towhee song bouts ($N_{Spotted\_towhee} = 796$ and $N_{Eastern\_towhee} = 796$). We tested the model on 697 song bouts, none of which were included in the training set. For the latter model, we separated the song bouts recorded in the zone of overlap from those in the non-overlap regions, and trained only on a subset of the non-overlap song bouts, retaining a sample size equal to that of the zone of overlap for testing. Once again, our training set was downsampled, such that we obtained a balanced subsample of Eastern towhees equal to that of Spotted towhees in our training set ($N_{Spotted\_towhee} = 796$ and $N_{Eastern\_towhee} = 796$). We tested this model on a sample of towhee songs from the non-overlap zones ($N = 216$), an equally sized sample of towhee songs from the zone of overlap ($N = 216$), and the sample of "hybrid/unsure" towhee songs ($N = 27$). For both models, we created a 10-fold cross-validation control and tuned the hyperparameters using a 10 x 16 grid using the caret package [39] in R, which allows us to evaluate every possible combination of the number of predictors to randomly sample at each split and the minimum number of samples needed to keep splitting nodes using Gini impurity to split nodes. We also repeated the random forest model using 100 different random seeds to assign recordings to the training and test sets, and we used these trials to estimate the overall accuracy of the classifier as well as a 95% confidence interval.

## Gradient boosting machine

We used a gradient boosting machine as another machine learning technique for classification of the song bouts. This technique is similar to random forest in that a large number of decision trees are trained on a subset of data and then tested on a withheld set of test data, but whereas these trees are independently generated with RFMs, a gradient boosting machine sequentially produces trees that are informed by the errors in previous trees. This technique can produce more accurate models, but has an increased risk of overfitting. As with the previous model, we obtained a random balanced subsample of 1592 song bouts ($N_{Spotted\_towhee} = 796$ and $N_{Eastern\_towhee} = 796$) to train the model and tested the model on 697 song bouts. The song feature data was centered and scaled. We used the 'xgb.train' function from the xgboost package [44] in R to train the model on the 16 song features using a multi-class softmax objective with early stopping (early_stopping_rounds = 10) to prevent overfitting. We set our model parameters to a maximum of 100 boosting iterations and used a multiclass log loss evaluation metric to calculate the classification error, such that the model attempts to improve each subsequent iteration based on how well the previous iteration is able to classify the observations.

## Deep learning model

Additionally, we used a deep learning model via the Torch package [45] in R that uses a multilayer perceptron (i.e., fully connected feedforward neural network) for classification of the song bouts. Again, we used a training set consisting of a random balanced subsample of 1592 song bouts ($N_{Spotted\_towhee} = 796$ and $N_{Eastern\_towhee} = 796$) and tested the model on 697 bouts. The data was centered and scaled prior to integration with the model. The network contains an input layer, two hidden layers, and an output layer. The input layer has a size of 16, defined by the number of song features. We used 64 neurons in the hidden layers, aiming to balance model performance and computational efficiency without overfitting. Each fully connected layer applies a linear transformation to the data, mapping it to a new space using the 'nn_linear' function. The first hidden layer maps the input features to the 64 hidden neurons, followed by a second hidden layer of the same size, both of which use the ReLU activation function to introduce non-linearity. The final fully connected layer outputs the model's raw predictions, which are then passed through a log-softmax activation function to compute class probabilities. The entire training dataset was passed through the model 100 times (num_epochs = 100) during the training process to allow the model to progressively learn from the training data to improve its accuracy and performance. Within each epoch, we used a small batch size of 32 for more stable convergence, such that 32 training samples were used per iteration so that the model frequently updated its parameters during the training process for more efficient training. We used the 'nnf_cross_entropy' loss function for the model to compute the unweighted cross entropy loss for the classification tasks and used an Adam optimization algorithm for Adaptive Moment Estimation with a 0.001 learning rate using the 'optim_adam' function to adjust the model's feature weights during training.

## Convolutional neural network using spectrogram images

In addition to the other machine learning models using song feature data, we also applied a convolutional neural network (CNN) on spectrogram images of the towhee recordings to test species classification using a different type of input data and computational approach. We modified an Audio Classification CNN model from [46]. To apply the model to our data, we first created spectrograms from.wav files using the librosa and matplot packages in python and applied low and high pass filters (high pass filter = 1000 Hz; low pass filter = 10000 Hz). Our model accepts images of 224x224 pixels with 3 color channels (RGB) in the input layer. The model then has a series of convolutional layers with 32, 128, 128, and 128 filters, each of size 3x3 pixels, which help to extract increasingly complex features from the input images. Each of the convolutional layers use ReLU activation functions to introduce non-linearity after each convolution. To reduce the dimensions of the feature maps, we use MaxPooling2D layers with a pool size of 2x2 after each convolutional layer to reduce the spatial dimensions of the feature maps, improving efficiency of the model and helping prevent

overfitting. The features are then flattened into a one-dimensional vector using the Flatten layer. This vector is then passed to a Dense layer with 1024 neurons and a ReLU activation for further abstraction and learning. The final output layer has 2 neurons (i.e., the Spotted and Eastern towhee classes), using the Softmax activation function to output class probabilities.

To ensure that the results would be comparable to our other models, we used the same training set, such that we trained the CNN on the same balanced set of towhee recordings ($N_{Spotted\_towhee} = 796$ and $N_{Eastern\_towhee} = 796$). We normalized our data by dividing the input data by 255 pixels and applied one-hot encoding on the labels. We trained the model using the 'model.fit' function in Python using a batch size of 32 to update the model's weights after each set of 32 samples, and we passed the entire training dataset through the network 10 times (epochs = 10). We also repeated this model in three different ways: 1) repeating this exact model 10 times, 2) using a filter size of 5x5 pixels and running that model 10 times, and 3) using a filter size of 7x7 pixels and running that model 10 times to capture a larger section of the images (Fig H in S1 Text).

### Genetic analysis

We downloaded fasta files of sequences of the cytochrome oxidase subunit I regions of the Spotted and Eastern towhee mitochondrial genomes available in the Barcode of Life Data Systems (BOLD) database and the NCBI database. We obtained a total of 18 Spotted towhee and 5 Eastern towhee sequences. We aligned these sequences using MAFFT [47], and we used a pairwise $F_{ST}$ (fixation index) in Arlequin [48] to quantify the level of genetic differentiation between the two species by measuring the proportion of genetic variation that is due to differences between populations. It also measures the extent to which individuals within populations are similar to one another, where a larger $F_{ST}$ describes a greater difference in allele frequencies within a population. We also did an Analysis of Molecular Variance (AMOVA) to quantify the proportion of genetic variation that is due to differences between versus within populations.

### PCA and Procrustes analysis of genetic data

Using the 'glPCA' function from the adegenet package [49] in R, we ran a PCA on the aligned fasta files from our genetic analysis. We followed with a Procrustes analysis to transform the two-dimensional plot of the first two principal components onto the target geographic plot of the longitude-latitude coordinates of the samples using the 'procrustes' function from the vegan package [41] in R. We then performed a resampling test using the 'protest' function in the vegan package to assess whether the real song data were more closely associated with geography than expected by chance.

### Assessing isolation by distance

We used a Mantel test to assess the correlation between pairwise distance of a geographic distance matrix and both a song feature distance matrix and genetic distance matrix. We computed a pairwise geographic distance matrix, with the distance between each pair of recording locations estimated by the 'distm' function from the geosphere package in R (geodesic distance 'distGeo' with WGS84 ellipsoid parameters). To obtain the song feature distance matrix of the raw song data, we used the 'dist' function from the stats package and calculated the Euclidean distance for each pairwise comparison of the 2785 song bout samples. We then performed a Mantel test of the geographic distance matrix versus each song feature distance matrix using the 'mantel' function (with parameters method = "spearman" and the default permutations = 999) from the vegan package.

We use the same distance estimation as above to obtain a geographic distance matrix for the sampling locations of the genetic sequences. Using the fasta file of the aligned genetic sequences from the Spotted and Eastern towhee mitochondrial genomes ($N_{Spotted} = 18$ and $N_{Eastern} = 5$), we compute a pairwise distance matrix using the 'dist.alignment' function from the seqinr package [50]. We then performed a Mantel test of the geographic distance matrix versus the genetic distance matrix using the same method as above.

## Data and code availability

Our data and code for all analyses described in this manuscript are available at https://github.com/CreanzaLab/TowheeAnalysis.

## Results

After gathering all song recordings of Eastern and Spotted towhees from the largest public repositories of natural sounds, Xeno-canto and Macaulay Library, and eliminating recordings that lacked song or had excessive background noise, we were able to analyze song bouts from 2785 individual towhees: 1718 song bouts from Eastern towhees and 1067 song bouts from Spotted towhees, in addition to 27 total song bouts that we categorized as "hybrid/unsure". For each song bout, we used the song-analysis software Chipper to segment the song into syllables and automatically extract 16 syllable and song features. In the Spotted towhee, several recorded birds had very rapid trills that could not be further parsed into syllables (i.e., there were no periods of silence within the trill) and, thus, were considered to be a single syllable.

From eBird records from 2013-2023, we estimated unique sightings of 203 hybrids; of these 203 putative hybrids, 190 had additional information to validate the hybrid identification, such as a photo, a description of plumage differences compared to either species, and/or a record that multiple birders observed and agreed with the identification. As we filtered out sightings from the same date, latitude, and longitude, we observed that most of the putative hybrids had been observed by more than one person (503 eBird records for 203 unique putative hybrids). The most common location for putative hybrids was Nebraska (109 unique sightings); in this state, there were unique sightings of 4,269 Eastern towhees (~2.6% hybridization) and 4,736 Spotted towhees (~2.3% hybridization) in the same time period. Our calculation of the hybridization rate is likely to be an underestimate due to underreporting of hybrids by birdwatchers [51]; our estimates are greater than other studies that use eBird sightings (0.064% average hybridization [51]) and less than focal studies of hybridizing populations in their zones of overlap (3–5% of Common and Thrush nightingales [52] and ~20% of Pied and Collared flycatchers [53]).

### Generalized Linear Models

We used generalized linear models (GLMs) to assess the degree to which longitude, latitude, species identification, and file conversion status contribute to the song feature trends we see in the Spotted and Eastern towhees (Table 1). Our analysis suggests that latitude is a significant predictor ($p < 0.05$) for nine song features, while longitude is a significant predictor of all song features except for average syllable frequency range. Species classification was significant in all song features except bout duration and rate of syllable production. Finally, we assessed whether converting the sampling rate or converting the file type to wav format for consistency had a meaningful effect on the song feature outcomes by including conversion status in the generalized linear models. The models provide evidence that the overall file conversion status was statistically significant only in five song features. For these five features, file conversion had similar t-values to other significant factors, suggesting that the effect of conversion on the song feature has a similar order of magnitude as other factors relative to their standard errors. However, the degree of significance was much less than that of the other variables (i.e., latitude, longitude, and species), indicating that it likely has less predictive power (Table 1). We see similar results when using only samples from the breeding season (Table B in S1 Text). We visualized songs before and after conversion to ensure that converting the sampling rate or file format of the recordings was not altering the timing or frequency of the songs. However, we did observe a geographic bias in file format, with mp3 files overrepresented in the west and thus in Spotted towhee recordings (Fig L in S1 Text), suggesting that conversion status is associated with the geographic variation in song features and is not itself a source of this variation.

When we plot these song features by longitude and species, the strongest predictors in our GLMs, we did not observe sharp discontinuities in song features in the zone of overlap between the two species (Fig E in S1 Text). The two species showed very similar distributions for some song features in the zone of overlap (e.g., bout duration and rate of syllable

**Table 1. Results of generalized linear models for relationship between song feature data and longitude, latitude, species classification, and file conversion status of Spotted and Eastern towhee.**

| Song Feature | Intercept t-value | Intercept p-value | Latitude t-value | Latitude p-value | Longitude t-value | Longitude p-value | Species t-value | Species p-value | Conversion t-value | Conversion p-value |
|---|---|---|---|---|---|---|---|---|---|---|
| Bout duration (ms) | 754.09 | p < 10⁻⁵⁰ | -5.08 | **4.08 x 10⁻⁷** | 12.50 | **6.46 x 10⁻³⁵** | -0.59 | 0.56 | -2.34 | **0.02** |
| Number of syllables | 93.29 | p < 10⁻⁵⁰ | -0.94 | 0.35 | 2.30 | **0.02** | 4.31 | **1.69 x 10⁻⁵** | -0.04 | 0.97 |
| Rate of syllable production (1/ms) | -161.93 | p < 10⁻⁵⁰ | -0.08 | 0.93 | -5.46 | **5.13 x 10⁻⁸** | 1.55 | 0.12 | -0.01 | 1.00 |
| Largest syllable duration (ms) | 197.84 | p < 10⁻⁵⁰ | 1.14 | 0.25 | -9.07 | **2.16 x 10⁻¹⁹** | -8.57 | **1.64 x 10⁻¹⁷** | 0.98 | 0.33 |
| Smallest syllable duration (ms) | 42.70 | p < 10⁻⁵⁰ | -4.68 | **3.06 x 10⁻⁶** | -12.79 | **1.92 x 10⁻³⁶** | -6.59 | **5.27 x 10⁻¹¹** | 1.89 | 0.06 |
| Average syllable duration (ms) | 92.06 | p < 10⁻⁵⁰ | -0.97 | 0.33 | -12.04 | **1.41 x 10⁻³²** | -6.73 | **2.09 x 10⁻¹¹** | 1.75 | 0.08 |
| Number of unique syllables | 78.52 | p < 10⁻⁵⁰ | -1.56 | 0.12 | 6.77 | **1.55 x 10⁻¹¹** | -7.11 | **1.49 x 10⁻¹²** | -2.65 | **0.01** |
| Degree of syllable repetition | 36.03 | p < 10⁻⁵⁰ | 1.72 | 0.09 | -3.31 | **9.38 x 10⁻⁴** | 5.81 | **6.91 x 10⁻⁹** | -0.75 | 0.46 |
| Average syllable upper frequency (Hz) | 1280.67 | p < 10⁻⁵⁰ | -4.56 | **5.41 x 10⁻⁶** | 3.44 | **5.93 x 10⁻⁴** | 15.85 | **p < 10⁻⁵⁰** | 0.16 | 0.87 |
| Average syllable lower frequency (Hz) | 979.40 | p < 10⁻⁵⁰ | 1.02 | 0.31 | 2.96 | **3.05 x 10⁻³** | 7.26 | **4.84 x 10⁻¹³** | -2.77 | **0.01** |
| Maximum syllable frequency (Hz) | 1176.66 | p < 10⁻⁵⁰ | -6.24 | **4.99 x 10⁻¹⁰** | 9.13 | **1.33 x 10⁻¹⁹** | 12.85 | **9.68 x 10⁻³⁷** | -1.71 | 0.09 |
| Minimum syllable frequency (Hz) | 934.93 | p < 10⁻⁵⁰ | 3.38 | **7.37 x 10⁻⁴** | -8.13 | **6.53 x 10⁻¹⁶** | 2.15 | **0.03** | -1.88 | 0.06 |
| Overall syllable frequency range (Hz) | 649.71 | p < 10⁻⁵⁰ | -7.26 | **4.97 x 10⁻¹³** | 11.67 | **9.50 x 10⁻³¹** | 11.15 | **2.70 x 10⁻²⁸** | -0.92 | 0.36 |
| Largest syllable frequency range (Hz) | 472.85 | p < 10⁻⁵⁰ | -5.38 | **7.93 x 10⁻⁸** | 4.27 | **2.02 x 10⁻⁵** | 8.46 | **4.21 x 10⁻¹⁷** | 1.77 | 0.08 |
| Smallest syllable frequency range (Hz) | 213.58 | p < 10⁻⁵⁰ | -4.44 | **9.54 x 10⁻⁶** | -5.75 | **1.01 x 10⁻⁸** | 7.17 | **9.55 x 10⁻¹³** | 3.22 | **1.32x10⁻³** |
| Average syllable frequency range (Hz) | 397.23 | p < 10⁻⁵⁰ | -5.03 | **5.16 x 10⁻⁷** | 0.85 | 0.40 | 9.35 | **1.69 x 10⁻²⁰** | 2.26 | **0.02** |

Values that are bolded and highlighted in green indicate p-values less than 0.05.

production), whereas for some frequency characteristics the two species had partially overlapping distributions and we observed a small cline in the zone of overlap (e.g., average syllable upper frequency). The number of unique syllables showed a clinal pattern in Spotted towhees, increasing in number of unique syllables from west to east, whereas the Eastern towhees had a consistent average number of unique syllables across their range (Fig E in S1 Text). This geographic variation in Spotted towhees could reflect either different selection pressures in the western portion of the range or cultural drift in which a single-syllable song rose to prevalence in the west.

## Linear Discriminant Analysis (LDA)

Our linear discriminant analysis of the raw song feature data was able to classify the species of the samples with 86.8% accuracy (balanced accuracy = 86.9%; Table 2). A permutation test suggests that the model performed significantly better than chance ($p < 10^{-3}$). We see similar results with song feature data from only the breeding season (Table C in S1 Text). Taking into account the agreement expected by random chance, Cohen's kappa suggests that there is substantial agreement between the predicted species classification and the actual classification of the samples ($\kappa = 0.73$; Table 2). The rate of syllable production had the highest contribution to the discriminant function, followed by the number of unique syllables. While there is some visible separation of species classification in one-dimensional space, there was still considerable overlap (Fig 3).

## Principal Components Analysis (PCA) and Procrustes analysis of song data

A PCA of the song data revealed considerable overlap in the song variation captured by PC1 and PC2. Individual songs in both the range of overlap and non-overlap were scattered throughout PC space; if the individuals in the zone of overlap had songs that showed greater species-level differentiation, we would expect that these points would have greater

**Table 2. Results for machine learning models for analyses of song feature data from song recordings of Spotted and Eastern towhees.**

| Analysis | Description | Overall Accuracy | Balanced Accuracy | Cohen's kappa (κ) | Permutation Test P-value (p) |
|---|---|---|---|---|---|
| Deep Learning | $N_{train}$=1592 (796 eastern; 796 spotted)<br>$N_{test}$=697<br>Raw song feature data was centered and scaled. | 0.900 | 0.906 | 0.794 | **p<10⁻³** |
| Gradient Boosting Machine | $N_{train}$=1592 (796 eastern; 796 spotted)<br>$N_{test}$=697<br>Raw song feature data was centered and scaled. | 0.882 | 0.888 | 0.758 | **p<10⁻³** |
| Random Forest Model 1 | $N_{train}$=1592 (796 eastern; 796 spotted)<br>$N_{test}$=697<br>Song feature data was log transformed. | 0.895 | 0.901 | 0.784 | **p<10⁻³** |
| Random Forest Model 2 | $N_{train}$=1592 (796 eastern; 796 spotted)<br>$N_{test\_nonoverlap}$=216<br>$N_{test\_overlap}$=216<br>Song feature data was log transformed.<br>Model was trained on a subset of bouts from the non-overlap region and tested on a subset of non-overlap bouts and all the samples from the overlap region. | overlap=0.843 non-overlap =0.935 | overlap=0.834 non-overlap =0.958 | overlap=0.678 non-overlap =0.865 | **overlap=p<10⁻³ non-overlap =p<10⁻³** |
| Convolutional Neural Network | $N_{train}$=1592 (796 eastern; 796 spotted)<br>$N_{test}$=697<br>Used 224x224 pixel images of song recording spectrograms. | 0.875 | 0.885 | 0.746 | **p<10⁻³** |
| Linear Discriminant Analysis | $N_{train}$=1592 (796 eastern; 796 spotted)<br>$N_{test}$=697<br>Used raw song feature data. | 0.868 | 0.869 | 0.726 | **p<10⁻³** |
| Linear Discriminant Analysis on Principal Components 1 & 2 | N=2785 (1718 eastern; 1067 spotted)<br>Song feature data was log transformed.<br>Data is also centered and scaled. | 0.762 | 0.740 | 0.488 | **p<10⁻³** |
| Uniform Manifold Approximation and Projection + Linear Discriminant Analysis | N=2785 (1718 eastern; 1067 spotted)<br>Song feature data was log transformed.<br>Data is also centered and scaled.<br>(results for each version of the UMAP of different "min_dist" and "n_neighbors" is in the R Script) | 0.864 | 0.858 | 0.713 | **p<10⁻³** |

P-values that are bolded and highlighted in green suggest statistical significance (α=0.05).

separation in PC space than those in the zone of non-overlap. PC1 explained 35.6% of the variance, with average syllable duration having the highest loading. PC2 explained 24.8% of the variance, with average syllable upper frequency having the highest loading. With a simple linear partition of this two-dimensional PC space (LDA), we could predict the species of a song with 76.2% accuracy. There was moderate agreement between the actual species classification and the model predictions beyond what is expected by chance (κ=0.49, permutation test $p<10^{-3}$; Table 2). A Procrustes analysis comparing the first two principal components of the song data to the latitude-longitude coordinates of the recording location indicated that, despite the extensive overlap between the songs of the two species in PC space, there is significantly more geographic signal in the song data than expected by random chance (Procrustes rotation 0.30, $p<0.001$, Fig 4A–4C).

## UMAP visualization

Projecting the song-feature data with a UMAP analysis showed that the songs of the Eastern and Spotted towhees did not form two species-specific clusters, but instead made one large cluster with Eastern towhee songs overrepresented on

PLOS Computational
Biology

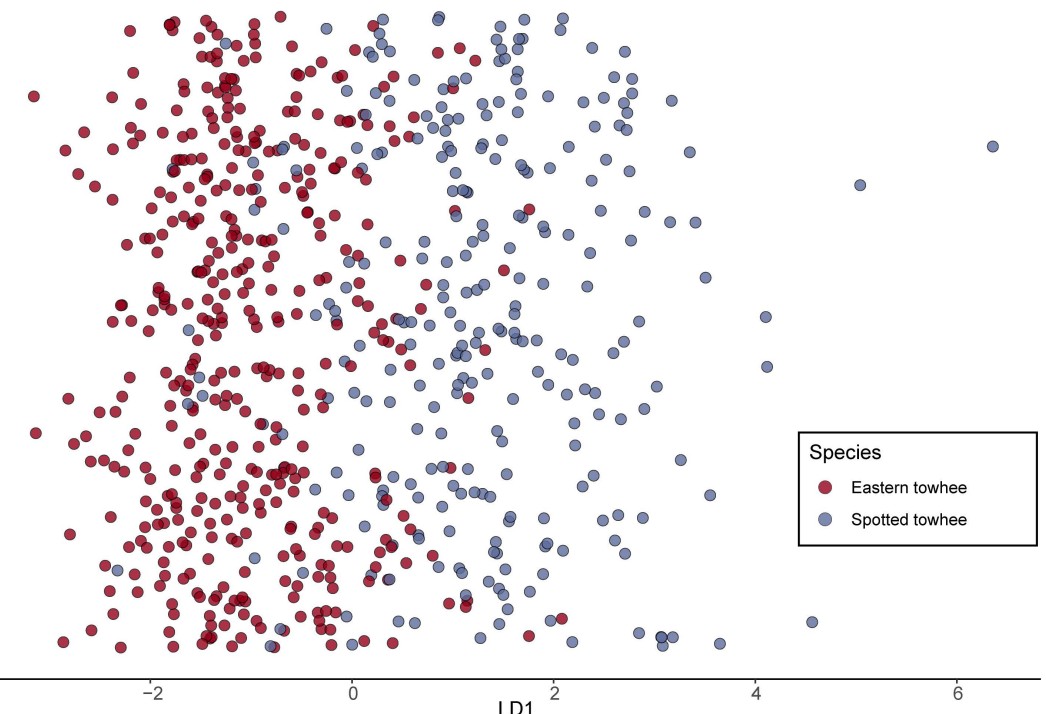

**Fig 3. Results of a Linear Discriminant Analysis trained on towhee song data.** Plot of LD1 results of a subset of towhee song bouts ($N_{test} = 697$) using a Linear Discriminant Analysis trained on raw song feature data from a balanced training set of Spotted and Eastern towhee bouts ($N_{train} = 1592$). The model revealed 86.8% prediction accuracy (balanced accuracy = 86.9%; Cohen's $\kappa = 0.73$). Points are jittered vertically for visualization.

one side and Spotted towhee songs on the other side (Fig 5); one much smaller cluster contained predominantly Spotted towhee songs. As in the PCA (Fig 4B), songs sampled from the zone of overlap were scattered throughout the UMAP projection. The species classification of each song could be better discriminated in the UMAP projection than the PCA plot; with a simple linear partitioning of the two-dimensional UMAP projection with n_neighbors = 15 and min_dist = 0.1, we could predict the species of a song with 86.4% accuracy (Table 2). We tested values of n_neighbors up to 50 and values of min_dist up to 0.9 and found similar prediction accuracies (75.3%–86.4%). This model performance showed substantial agreement with the actual species classification, significantly better than what is expected from random chance (permutation test $p < 10^{-3}$; $\kappa = 0.71$). The "hybrid/unsure" recordings were located throughout the large cluster of the UMAP projection (Fig 5). The same analysis using data from samples recorded only during the breeding season showed similar results (Fig F in S1 Text). In general, the location of points in a UMAP projection are not stable between runs and are influenced by parameters set by the user, so we do not interpret the distance between points or clusters in this projection to be meaningfully linked to song variation [51,52]; however, we did observe that the relative locations of the song-feature data on the UMAP projection showed an association with the species of the recording (Fig 5).

## Random forest model

Our random forest model trained on samples from the entire geographic range had an accuracy of 89.5% ($\kappa = 0.78$; permutation test $p < 10^{-3}$) when tested on a subset of song samples that had been withheld from training (Fig 6A and Table 2). The most important feature in the decision trees was the largest syllable duration. For each recording in the test set, we estimated the confidence of the classifier by tabulating the fraction of trees in the random forest that supported the correct classification (Fig 6B); these confidence values tended to decrease toward the zone of overlap, suggesting that the songs

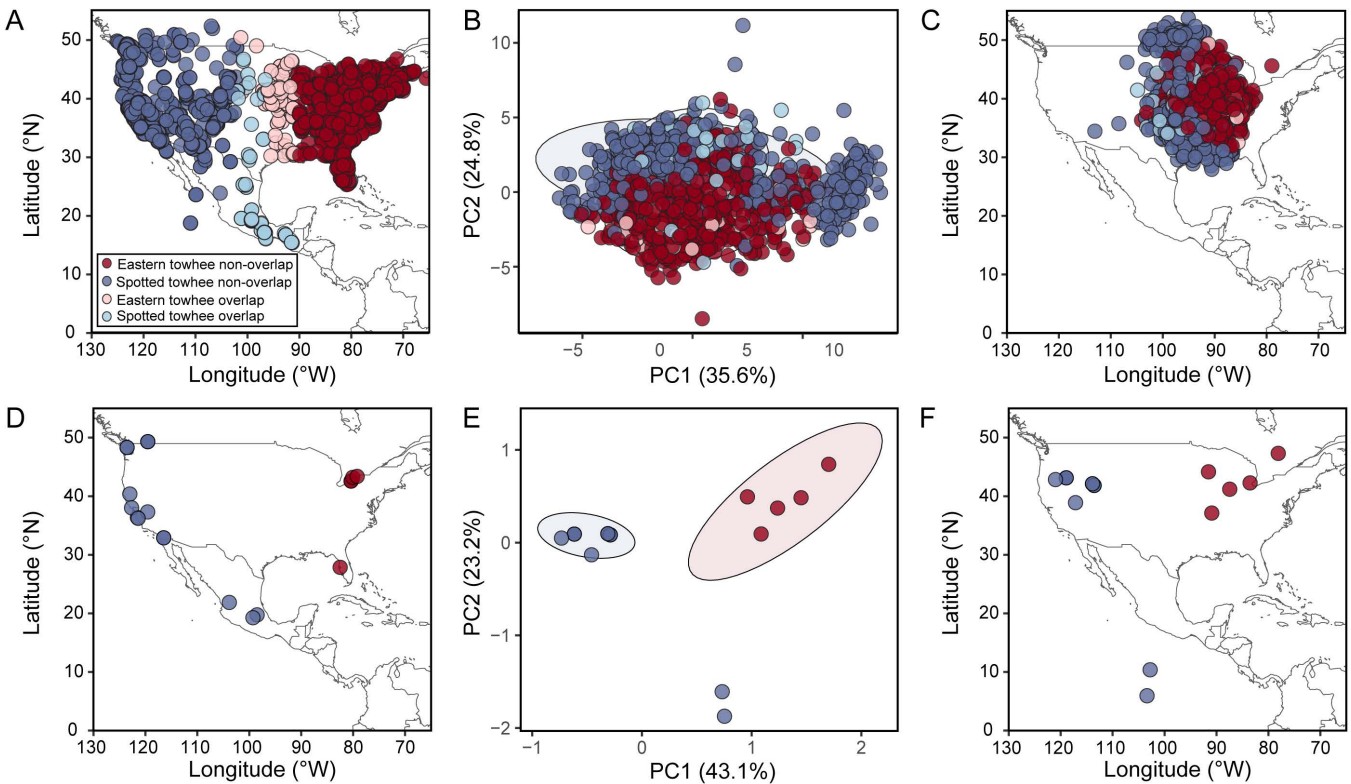

**Fig 4. Spatial distribution of Spotted towhee and Eastern towhee song data and genetic data.** Song data is shown in panels **A–C** ($N_{total\_recordings}$ = 2785; $N_{Spotted\_towhee}$ = 1067; $N_{Eastern\_towhee}$ = 1718) and genetic data is shown in panels **D–F** ($N_{total}$ = 23; $N_{Spotted\_towhee}$ = 18; $N_{Eastern\_towhee}$ = 5). **(A)** Distribution of song recordings in North America. **(B)** Principal component analysis (PCA) of song bouts using 16 song features. **(C)** Procrustes analysis of song data using PC1 and PC2 from the PCA in panel **B**. **(D)** Distribution of genetic sequences obtained from the Barcode of Life Data Systems database and NCBI. **(E)** PCA using single nucleotide polymorphisms of aligned sequences of the cytochrome oxidase subunit I regions of the mitochondrial genome. **(F)** Procrustes analysis of genetic data using PC1 and PC2 from the PCA analysis in panel **E**. Ellipses indicate 95% confidence intervals. Base maps (panels A, C, D, and F) were made with Natural Earth (http://www.naturalearthdata.com/).

in the zone of overlap were more challenging to classify. We repeated this analysis using 100 different subsets as training data, and the average accuracy was 89.1%±0.2.

Our model trained on the samples from the zone of only non-overlap was tested on all samples from the zone of overlap (216 song bouts) and on a withheld subset of the same number of samples from the zone of non-overlap (216 song bouts) which had an accuracy of 84.3% ($\kappa$=0.68; permutation test $p<10^{-3}$; accuracy$_{100seeds}$=83.7%±0.2) and 93.5% ($\kappa$=0.87; permutation test $p<10^{-3}$; accuracy$_{100seeds}$=89.8%±0.4), respectively (Fig 6C and Table 2). The most important feature was the largest syllable duration. As in Fig 6B, the classifier's confidence in its species predictions tended to decrease toward the zone of overlap (Fig 6D). When we used the model trained on only the samples from the zone of non-overlap to predict the species classification of the samples from "hybrid/unsure" towhee recordings, it predicted 16 Spotted towhees and 11 Eastern towhees (Fig 6E). Additionally, we tested the models using both 100 and 1000 trees and on only the recordings from the breeding season and obtained similar results (Tables C–D and Fig G in S1 Text).

### Gradient boosting machine

Our gradient boosting machine model was able to predict species classification with 88.2% accuracy ($\kappa$=0.76; permutation test $p<10^{-3}$; Table 2). The most important feature in the gradient boosting machine was the number of unique

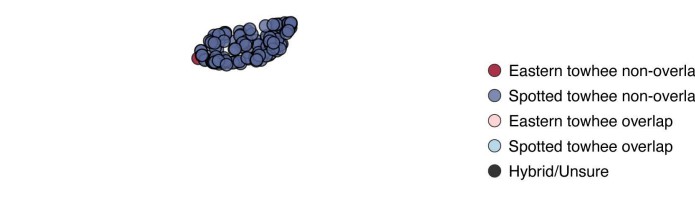
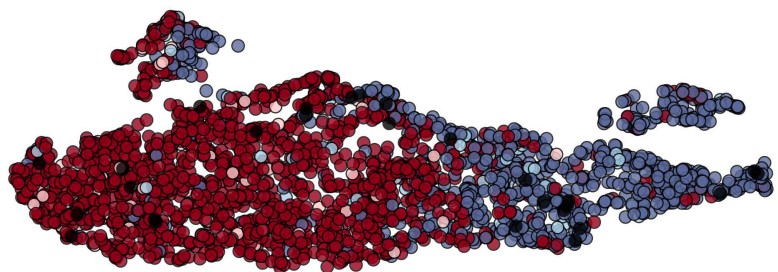

Eastern towhee non-overlap
Spotted towhee non-overlap
Eastern towhee overlap
Spotted towhee overlap
Hybrid/Unsure

**Fig 5. UMAP projection of Eastern and Spotted towhee song-feature data.** Each point represents an analyzed song bout ($N_{total\_bouts} = 2785$; $N_{Spotted\_towhee} = 1067$; $N_{Eastern\_towhee} = 1718$), with Eastern towhee songs shown in shades of red and Spotted towhee songs in shades of blue. The lighter colors represent recordings from the zone of species overlap. Black dots indicate the 27 recordings from individuals that were classified as potential hybrids ("hybrid/unsure").

syllables, contributing about 25% to the model's accuracy. Similar to the random forest models, the second most important feature was largest syllable duration, contributing about 18%.

### Deep learning model

Out of all our machine learning algorithms, our deep learning algorithm was able to predict the species classification with the highest accuracy of 90.0% and showed substantial agreement between the predicted classifications and the actual species classification, performing better than expected by chance ($\kappa = 0.79$; permutation test $p < 10^{-3}$; Table 2). Nonetheless, all our machine learning classifiers had similar prediction accuracies (Tables 2 and C in S1 Text) despite using different methods. Additionally, 43.4% of the incorrect predictions from the random forest model 1, gradient boosting machine, and deep learning algorithm were misclassified across all three models.

### Convolutional neural network

Using a convolutional neural network on spectrogram images of the towhee recordings as an alternative approach to assessing species classification, we obtained an accuracy of 87.52% (balanced accuracy = 88.51%; $\kappa = 0.7457$; permutation test $p < 10^{-3}$), which is similar to the other machine learning algorithms we used in this manuscript (Table 2). We also ran this CNN model 1) 10 times using the same 3x3 filter, 2) 10 times using a 5x5 filter size, and 3) 10 times using a 7x7 filter size, obtaining an average accuracy of 88.32% ± 1.02%, 90.06% ± 1.52%, and 89.74% ± 1.07%, respectively (Fig H in S1 Text).

### Mantel test

A Mantel test of song feature distance versus geographic distance showed that all song features had a weak positive correlation with geographic distance ($p < 0.001$; Table 3). Even song features that had an extremely weak correlation with geographic distance (e.g., maximum syllable frequency) were found to be more closely associated with geography than the randomly permuted matrices (r=0.018, $p < 0.001$).

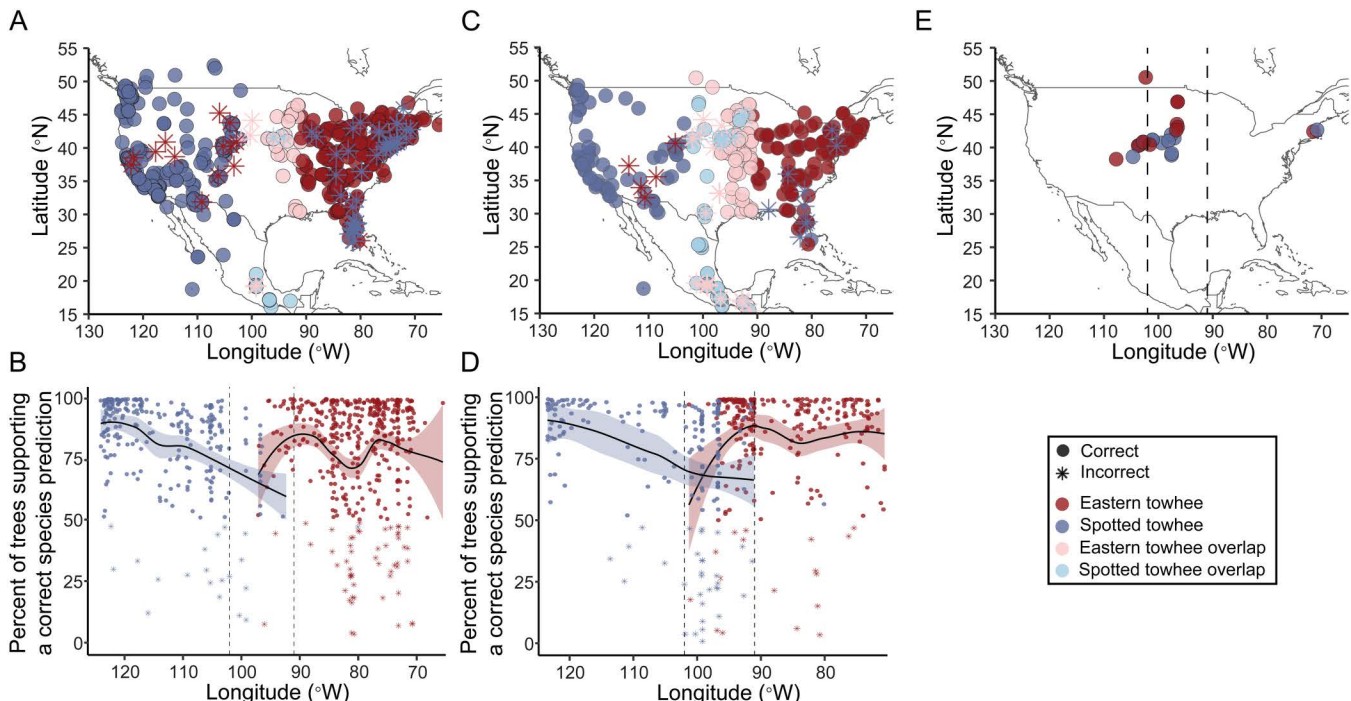

**Fig 6. Geographic distribution of random forest model predictions of Spotted and Eastern towhee species identity.** Predictions were based on a model trained on 16 song features from samples of Spotted towhees and Eastern towhees. **(A)** We trained a model on song data from the entire geographic range of both species ($N_{Spotted\_towhee} = 796$; $N_{Eastern\_towhee} = 796$) and tested how well it predicted the species identification of a subset of all song samples ($N_{test} = 697$; accuracy = 89.5%). **(B)** The same results from **(A)** are plotted based on the percent of trees in the random forest classifier that supported the correct species identification; when less than 50% of trees supported the correct identification, the classifier made an incorrect species prediction. The average confidence in species classifications tended to decrease toward the zone of overlap for both species. **(C)** We then trained a second model on a subset of samples obtained only from the non-overlap zone ($N_{Spotted\_towhee} = 796$; $N_{Eastern\_towhee} = 796$) and tested it on a random subsample of song bouts from both the zone of non-overlap ($N_{test\_nonoverlap} = 216$; accuracy = 93.5%) and the zone of overlap (i.e., 102°W - 91°W; $N_{test\_overlap} = 216$; accuracy = 84.3%). **(D)** The same results from **(C)** are plotted based on the percent of trees in the random forest classifier that supported the correct species identification; again, the average confidence of species identifications tended to decrease toward the zone of overlap for both species. **(E)** We used the same model from panels **C** and **D** to predict species identity of song bouts from recordings of "hybrid/unsure" towhees ($N_{predict} = 27$). The model predicted that 16 of these "hybrid/unsure" recordings were Spotted towhees and 11 were Eastern towhees, with no discernable longitudinal gradient in the predictions. The dotted line represents the zone of overlap determined by the co-occurence of Eastern towhee and Spotted towhee song recordings (102°W - 91°W). Base maps (panels A, C, and E) were made with Natural Earth (http://www.naturalearthdata.com/).

## PCA and Procrustes analysis of genetic data

We analyzed the limited genetic data available for the most commonly sequenced gene in the Spotted and Eastern towhees, obtaining 23 cytochrome oxidase I sequences ($N_{Spotted\_towhee} = 18$; $N_{Eastern\_towhee} = 5$). A PCA using these mtDNA sequences revealed 3 different clusters. Eastern towhees formed their own cluster, while the Spotted towhees were split into two clusters: one group that contained two individuals from Mexico and a second group that included all individuals from the United States and Canada, along with one individual from Mexico. Eastern and Spotted towhees cluster separately, suggesting that they are genetically different, at least based on SNPs from COI mtDNA sequences. PC1 explained 43.1% of the variance in the genetic data, and PC2 explained 23.2%. A Procrustes analysis comparing the first two principal components of the genetic data to the latitude-longitude coordinates of the sampling location indicated that there is significant geographic signal in the genetic data (Procrustes rotation 0.85, $p < 0.001$, Fig 4D–4F).

**Table 3. Results for Mantel tests of each song feature distance matrix versus geographic distance matrix.**

| Song Feature | Mantel Statistic (r) | P-Value (p) |
|---|---|---|
| Bout duration (ms) | 0.267 | <0.001 |
| Number of syllables | 0.183 | <0.001 |
| Rate of syllable production (1/ms) | 0.282 | <0.001 |
| Largest syllable duration (ms) | 0.234 | <0.001 |
| Smallest syllable duration (ms) | 0.102 | <0.001 |
| Average syllable duration (ms) | 0.165 | <0.001 |
| Number of unique syllables | 0.198 | <0.001 |
| Degree of syllable repetition | 0.284 | <0.001 |
| Average syllable upper frequency (Hz) | 0.192 | <0.001 |
| Average syllable lower frequency (Hz) | 0.069 | <0.001 |
| Maximum syllable frequency (Hz) | 0.018 | <0.001 |
| Minimum syllable frequency (Hz) | 0.187 | <0.001 |
| Overall syllable frequency range (Hz) | 0.033 | <0.001 |
| Largest syllable frequency range (Hz) | 0.037 | <0.001 |
| Smallest syllable frequency range (Hz) | 0.260 | <0.001 |
| Average syllable frequency range (Hz) | 0.125 | <0.001 |

A Mantel test of genetic distance versus geographic distance among Spotted and Eastern towhees showed a moderately strong positive correlation (r=0.6361; p<0.001), such that samples that were further away from one another had more genetic differences.

## Genetic analysis

Our genetic analysis revealed a pairwise $F_{ST}$ value of 0.64, with an AMOVA showing that 63% of the variance was between species and 37% of the variance was within species. Although this analysis was performed only on the 23 available sequences, our AMOVA results broadly agree with the genetic PCA analysis, where we see a separate cluster for the Eastern towhees (Fig 4E).

## Discussion

Since oscine songbirds learn their songs from conspecifics, they are an excellent model system for studying how culturally transmitted traits affect evolution and speciation. Here, we examine variation in the learned songs of the Spotted and Eastern towhees, a sister species pair that each have broad ranges, together covering ~10 million $km^2$. Since we have access to song recordings sampled throughout North America, we can detect geographic variation that exists at these continental scales. This pair of sister species has a zone of geographic overlap that is smaller than their respective allopatric ranges, where it has been noted that individuals occasionally hybridize (Fig B in S1 Text). Although hybridization occurs between these sister species—using eBird data, we estimate the highest density of hybrids to occur in Nebraska, comprising ~2.3% to 2.6% of towhee sightings in the state—our mtDNA analysis suggested levels of genetic differentiation similar to those of other distinct avian species [54], indicating the potential for reproductive isolation in these sister species. However, we had few genotyped samples, and the majority of them were from the outer edges of the ranges, which limits our ability to assess genetic differentiation in the overlap zone. Nonetheless, Spotted and Eastern towhees are considered separate species due to genetic, morphological, and song differences and seem to maintain themselves as separate species. Therefore, individuals are likely using specific traits or a combination of traits to recognize conspecifics, particularly because variation in song features exists not only between sister species but also within species. In this study, we analyze song features that might differ between the Spotted and Eastern towhees, and we investigate whether differences in song features could potentially allow individuals to reliably distinguish species-specific songs.

Our statistical analysis showed substantial variation in song features across the longitude and latitude gradient overall (Figs C–E in S1 Text), but longitude was a significant predictor for more song features than latitude (Table 1), which is consistent with the geographic distribution of the two species at similar latitudes but mostly different longitudes. Additionally, the populations of Spotted towhees in the westernmost edge of the geographic range appear to have larger variation in some of the song features, including number of syllables, average syllable duration, and degree of syllable repetition (Figs C–E in S1 Text); this pattern appears to correspond to the region where Spotted towhees have more rapid trills that occasionally were performed as a single syllable (with no separation between the repeated elements). A similar study in a sister-species pair of suboscine *Pyriglena* fire-eye antbirds found that song variation increased at the contact zone where introgression occurs [55]. We do not see this same pattern in our study; the degree of song feature variation at the zone of range overlap is either consistent with (e.g., smallest syllable frequency range) or less than the amount of variation found at the edges, particularly when compared to the westernmost region of the Spotted towhee range (e.g., average syllable duration; Fig E in S1 Text). Perhaps, since the Spotted and Eastern towhees learn their songs, unlike the suboscines, the pattern of variation at the hybrid zone may be a result of both conspecific and heterospecific copying as opposed to purely genetic introgression. Further, the maintenance or reduction of variation at the zone of overlap could be caused by positive frequency-dependent cultural selection, where the more common songs in the local area are favored (e.g., [56]). As individuals traverse the zone of overlap, they are exposed to more heterospecific songs and fewer conspecific songs, while simultaneously being exposed to fewer songs from the edges of the species' ranges, potentially leading to individuals learning elements of the songs that are more common in that area. This pattern of learning could lead to reduced song distinguishability between the species in the region where song differences would be most useful for assortative mating. Together, our results suggest that Spotted towhee songs differ predominantly in the westernmost part of their range, not near the zone of species overlap, and geographic variation alone does not account for the song variation between Spotted and Eastern towhees.

Overall, we found that no single song feature can be used to reliably distinguish between species' songs. The PCA and UMAP plots revealed minimal separation in song variation between the two species (Figs 4B and 5). However, the song features of the two species were biased toward different portions of the cluster, such that we could divide each plot into two sections and discriminate species with 76.2% accuracy for the PCA and 86.4% accuracy for the UMAP. A Procrustes analysis showed weak but significant geographic structure in the distribution of song data (Fig 4C). Performing with even greater accuracy, our random forest classifier was able to correctly predict the species of 89.5% of songs, suggesting that several song features, in combination with one another or with morphology, could allow individual birds to reliably distinguish members of their own species. However, our random forest models trained on samples from the edges of the range were able to predict the species identification of songs from outside the zone of overlap (~93.5%) better than those inside the zone of overlap (~84.5%) (Fig 6C and Table C in S1 Text), supporting the notion that intermediate songs seem to exist [29]. Additionally, our convolutional neural network trained on spectrogram images of the Spotted and Eastern towhee song recordings had similar prediction probabilities as the other machine learning algorithms, suggesting that even when using a different type of classification approach (i.e., raw pixel data versus extracted song features), we obtain similar results. Overall, our machine learning classifiers showed ~85–90% accuracy in predicting the species of towhee songs, which is significantly better than chance; however, the remaining 10–15% of Eastern and Spotted towhee songs did not have readily distinguishable song features. With the relatively low rate of hybridization between these two species, birds are likely using other species recognition cues in addition to the song features we measured, but these intermediate songs might still play a role in heterospecific mating.

Our observation that songs of the Spotted and Eastern towhee are less distinguishable in the zone of overlap than at the edges of the ranges does not support the hypothesis that this sister-species pair has developed increased song differentiation in this zone of overlap to prevent hybridization; this reduced distinguishability could lead to some level of heterospecific mating and thus to incomplete reproductive isolation. Further, the classifier tended to have less confidence

in its predictions in the zone of overlap, suggesting that even correctly identified songs were more difficult to classify in this region. However, incorrect predictions were distributed relatively evenly across the geographic range and not concentrated in the zone of overlap. Still, while our machine learning algorithms can correctly predict the species assigned to the recording at a much higher accuracy than random chance, it is unknown whether some of the songs that were correctly predicted in our model actually come from genetically admixed individuals who have learned the song of one species and were not identified by the recordist as a hybrid. For example, a hybrid individual located at the zone of overlap may have learned its entire song from a non-hybrid parent or other adult of either species, so the species classification of an individual based on song may not accurately reflect its genotype. In addition, the "hybrid/unsure" recordings—songs of potential hybrids and of towhees that were difficult for the recordist to categorize as Eastern or Spotted—did not fit a predictable longitude-based pattern of categorization by the random forest classifier. A larger sample of "hybrid/unsure" recordings accompanied by more extensive morphological descriptions could provide us with greater insight into the pattern of song distinguishability that may exist at the hybrid zone.

To compare geographic distance and song variation, we used a Mantel test to assess whether songs accumulate changes by cultural drift, fitting an isolation-by-distance pattern. Our analysis showed that geographic distance and song feature distance tend to increase together; however, the relationship between the two matrices was weak. This suggests that isolation by distance, at least on a large scale, does not fully explain the song differences we see in the Spotted and Eastern towhees. Future studies incorporating environmental characteristics at the population level would give us greater insight into factors affecting the evolution and speciation of avian populations.

Altogether, our analyses suggest that reinforcement of species boundaries is not readily detectable in towhee song, and other factors, such as cultural drift or differing habitats (e.g., vegetation density, altitude, climate), might also help explain the extensive amount of variation that we see in song features within and between species. For example, the acoustic adaptation hypothesis suggests that birdsong evolves under the constraints of the sound transmission properties of a given environment, and the songs are structured to increase the fidelity of song transmission in the native environment [57–59]. As such, individuals with certain song traits, particularly frequency bandwidth [60], may be more fit in specific environments if their song is able to transmit through the environment and be more easily detected. Additionally, as individuals traverse the zone of overlap into the other species' range, female preference for certain traits could lead to differential fitness in males depending on their songs [5]. For example, if females prefer mates with traits that fall towards the extremes, those with intermediate songs would have decreased fitness. This may contribute to constraining the width of the hybrid zone and maintaining reproductive isolation.

Future studies using playback experiments in the field could assess how individuals respond to conspecific versus heterospecific song, which would more directly test whether individuals can discriminate between songs and whether a preference exists for certain song features. For example, a study of the squamate antbird (*Myrmoderus squamosus*) and the white-bibbed antbird (*Myrmoderus loricatus*), a sister-species pair of suboscine birds which have extensive overlap in song variation but negligible genetic introgression, used song-playback experiments of both conspecific and heterospecific songs to assess whether these species exhibited vocal recognition; this study showed that even small differences in song seem to play a role in competitor recognition and mate attraction, suggesting that species recognition could evolve even with little vocal divergence [61]. While our study does not incorporate playback experiments, we find similar results showing that even though Eastern and Spotted towhees have a large amount of song-feature overlap (Figs 3–5), subtle song differences between the two species can lead to relatively reliable species distinctions that are detectable with a machine-learning classifier trained on multiple song features but not with simple statistical comparison of single song features. However, it is interesting to note that these distinctions are less apparent in the zone of species overlap, where they have been hypothesized to be most useful. Based on these results, we hypothesize that there is not a strong selection pressure in the zone of overlap favoring song differentiation to limit hybridization below its current level. Perhaps since this zone of overlap is relatively sparsely populated with both Spotted and Eastern towhees, the limited opportunity

for potential breeding interactions between species does not lead to a strong advantage for male birds that have reliably distinguishable songs or for the female birds that prefer them.

Further, differences in the trills of the Spotted and Eastern towhee may be features that are important for species recognition. A study by Richards [62] suggests that the trills of the northeast populations of Eastern towhees function as a messaging component and may indicate species recognition. The invariant-features hypothesis suggests that song features that vary less within a species are likely to be features that are used in species recognition [63]. While none of our measured song features could accurately predict species classification when averaging over an entire song bout, it is possible that features of the trill could provide more species-specific information. Therefore, future analyses could compare only the trills of Spotted and Eastern towhee songs to determine whether these trills have more species-specific differences than the songs as a whole.

Since we used two different public repositories of bird song recordings, files had various sampling rates and formats (wav, mp3, and m4a). Because of the potential effect of file type and sampling rate conversion on our analysis, we assessed whether overall conversion status would show statistical significance in our GLM. While the file conversion status had a similar magnitude of effect on the song features as other factors, the degree of significance was much less than the other significant factors in the GLM (Table 1). To investigate the factors underlying this effect, we first verified that the frequency and timing of individual songs did not change after we converted the format or sampling rate; then, we assessed whether the converted files were evenly distributed across our sample, and we found that our database had a higher proportion of Eastern towhee songs in wav format and Spotted towhee songs in mp3 format (Fig L in S1 Text). Since the song features we studied show significant geographic and species differences, the influence of conversion status in our GLM appears to be related to this biased sampling and not to an effect of conversion on the song features. Further, in addition to the machine learning models reported here, we also ran each model again incorporating overall conversion status as a feature. We found that the changes are very minimal, with a maximum change in accuracy of only 1.2% (see Supplemental Methods in S1 Text). Conversion status was also the feature of lowest importance or second low-est importance in the PCA, GBM, and both RFMs. When color-coding our LDA, PCA, and UMAP plots by overall conversion status, we saw significant overlap with no apparent separation between groups (Figs I – K in S1 Text). Together, this suggests that the overall conversion status does not seem to be useful in making species predictions using our machine learning models.

Since we analyzed public repositories of bird song recordings, we are limited in our ability to make direct links between genetic data, song data, and morphology of individuals. We do not know the actual degree of genetic admixture of any individual in our analysis or whether the degree of admixture correlates with song differences in individuals. Further, citizen scientists are likely to consider a bird's sound and location in their identification, possibly without actually seeing a bird's plumage and almost certainly without any genetic data, so individuals in our analyses may not all be accurately identified. However, using these repositories significantly increases the amount and geographic coverage of data we can analyze, allowing for a larger-scale analysis than field research efforts. While our study does not make direct links between genetics, morphology, and song at the individual level, it sheds light on the potential patterns that may exist in the evolution of learned song in oscine birds. We hope this study provokes further research on song variation linked to genotype and phenotype data at the contact zone to better understand the potential selection pressures in effect.

## Supporting information

**S1 Text. The role of learned song in the evolution and speciation of Eastern and Spotted towhees. Table A. Additional information about "hybrid/unsure" recordings.** Information provided in Macaulay Library and eBird by recordists to accompany recordings in which the observer suspected a hybrid Eastern × Spotted towhee or was unsure whether the bird was an Eastern or Spotted towhee. **Table B. Associations between song feature data and recording metadata.** Results for generalized linear models of song feature data against latitude, longitude, species classification,

and conversion status (determined by whether the audio file type and/or sampling rate was changed from the original song file). These analyses include only recordings from breeding season months. P-values that are bolded and highlighted in green are below an α = 0.05 threshold. **Table C. Results for machine learning models for analyses of song feature data from the subset of recordings of Spotted and Eastern towhees that were obtained during the breeding season.** P-values that are bolded and highlighted in green are below α = 0.05. **Table D. Prediction accuracies of random forest models trained on 16 song features from samples of Spotted towhees and Eastern towhees using different numbers of decision trees.** (A-C) Predictions of subset of all song samples ($N_{test}$ = 697) trained on song data from the entire geographic range ($N_{Spotted\_towhee}$ = 796; $N_{Eastern\_towhee}$ = 796). (D-F) Models trained on a subset of samples obtained from the non-overlap zone ($N_{Spotted\_towhee}$ = 796; $N_{Eastern\_towhee}$ = 796). The model was tested on a random subsample of song bouts from both the zone of non-overlap ($N_{test\_nonoverlap}$ = 216) and the zone of overlap ($N_{test\_overlap}$ = 216). Increasing the number of trees did not substantially change the accuracy of the models' predictions. We report the results from models with 500 trees in the main text. **Fig A. Frequency of song recordings of each species by longitude.** Line plot of the fraction of Spotted towhee and Eastern towhee song recordings, calculated as the number of recordings of each species divided by the total number of recordings of either species across North America during the breeding season ($N_{total}$ = 2785; $N_{Spotted\_towhee}$ = 1067; $N_{Eastern\_towhee}$ = 1718). The black dashed vertical lines represent the zone of song overlap, determined based on the co-occurence of song recordings (102°W – 91°W). **Fig B. Map of putative Spotted towhee × Eastern towhee hybrid sightings (N = 203) during the breeding season.** 190 of the 203 hybrid sightings in the eBird database had either more than one observer, available media, or observer comments. The dotted line represents the zone of overlap determined by the co-occurence of song recordings (102°W – 91°W); 77.8% of these putative hybrid sightings fall within this zone of song overlap. The sightings of putative hybrids are shifted west compared to the zone of song overlap, potentially suggesting that hybridization is more likely when Eastern towhees are the rarer species. Sighting data obtained from eBird; metadata available at https://github.com/CreanzaLab/TowheeAnalysis. Base map made with Natural Earth (http://www.naturalearthdata.com/). **Fig C. Maps of song features related to duration and number of syllables, plotted by their recording location.** Each point represents an analyzed song bout ($N_{total}$ = 2788; $N_{Spotted\_towhee}$ = 1069; $N_{Eastern\_towhee}$ = 1719). The color scale on the bottom left of the map corresponds to the log-transformed song feature value of each recording. The shape of each point indicates the species classification of the recording. Base maps made with Natural Earth (http://www.naturalearthdata.com/). **Fig D. Maps of song features related to frequency, plotted by their recording location.** Each point represents an analyzed song bout ($N_{total}$ = 2788; $N_{Spotted\_towhee}$ = 1069; $N_{Eastern\_towhee}$ = 1719). The color scale on the bottom left of the map corresponds to the log-transformed song feature value of each recording. The shape of each point indicates the species classification of the recording. Base maps made with Natural Earth (http://www.naturalearthdata.com/). **Fig E. Log-transformed song features plotted based on the longitude of the recording location.** Each point represents an analyzed song bout ($N_{total}$ = 2788; $N_{Spotted\_towhee}$ = 1069; $N_{Eastern\_towhee}$ = 1719), with Eastern towhee songs shown in shades of red and Spotted towhee songs in shades of blue. The lighter colors represent recordings from the zone of species overlap. The black line represents a smoothed average using a generalized additive model-fitting method. **Fig F. UMAP projection of Eastern and Spotted towhee song-feature data using the subset of samples recorded during the breeding season.** Each point represents an analyzed song bout ($N_{total\_bouts}$ = 2436; $N_{Spotted\_towhee}$ = 874; $N_{Eastern\_towhee}$ = 1562), with Eastern towhee songs shown in shades of red and Spotted towhee songs in shades of blue. The lighter colors represent recordings from the zone of species overlap. Black dots indicate the 26 recordings from individuals that were classified as potential hybrids ("hybrid/unsure"). Using a linear discriminant classifier to partition the UMAP projection into two sections, we could accurately predict the species of 84.9% of recordings (n_neighbors = 15 and min_dist = 0.1). Using values of n_neighbors up to 50 and values of min_dist up to 0.9, the linear classifier showed prediction accuracies ranging from 81.3% to 86.4%. **Fig G. Geographic distribution of species predictions using a random forest model trained on 16 song features from only those samples of Spotted towhees and Eastern towhees that were recorded during the breeding season.** (A) We trained a model on song data

from the entire geographic range of songs recorded during the breeding season ($N_{Spotted\_towhee}=652$; $N_{Eastern\_towhee}=652$) and tested how well it predicted the species identification of a subset of breeding-season song samples ($N_{test}=609$; accuracy=92.4%). (B) We then trained a second model on a subset of samples obtained during the breeding season from the non-overlap zone ($N_{Spotted\_towhee}=652$; $N_{Eastern\_towhee}=652$). This model was tested on a random subsample of breeding-season song bouts from both the zone of non-overlap ($N_{test\_nonoverlap}=167$; accuracy=92.8%) and the zone of overlap ($N_{test\_overlap}=167$; accuracy=87.4%). (C) We used the same model from panel B to predict species identity of song bouts from recordings of "hybrid/unsure" towhees ($N_{predict}=26$). The model predicted that 16 of these "hybrid/unsure" recordings were Spotted towhees and 10 were Eastern towhees, with no discernable longitudinal gradient in the predictions. The dotted line represents the zone of overlap determined by the co-occurence of Eastern towhee and Spotted towhee song recordings (102°W - 91°W). Base maps made with Natural Earth (http://www.naturalearthdata.com/). **Fig H. Validation accuracies of 10 runs for a convolutional neural network trained on spectrogram images of song recordings of Spotted towhees and Eastern towhees.** We trained a model on spectrogram images of song recordings from the entire geographic range of both species ($N_{Spotted\_towhee}=796$; $N_{Eastern\_towhee}=796$) and tested how well it predicted the species identification of a subset of all song samples ($N_{test}=697$). We passed the entire training dataset through the network 10 times using (A) 3x3 pixel filters (average accuracy of last epoch=88.32% ± 1.02%), (B) 5x5 pixel filters (average accuracy of last epoch=90.06% ± 1.52%), and (C) 7x7 pixel filters (average accuracy of last epoch=89.74% ± 1.07%). **Fig I. Principal component analysis of song bouts using 16 song features.** Each point represents an analyzed song bout ($N_{total\_recordings}=2785$; $N_{Spotted\_towhee}=1067$; $N_{Eastern\_towhee}=1718$), color-coded by conversion status. 'Converted' denotes samples in which the original recording file was converted to a.wav file (from.mp3 or.m4a) and/or the sampling rate was converted to 44,100 Hertz. Ellipses indicate 95% confidence intervals. **Fig J. UMAP projection of Eastern and Spotted towhee song-feature data.** Each point represents an analyzed song bout ($N_{total\_bouts}=2785$; $N_{Spotted\_towhee}=1067$; $N_{Eastern\_towhee}=1718$; $N_{hybrid/unsure}=27$), color-coded by overall conversion status. 'Converted' denotes samples in which the original recording file was converted to a.wav file (from.mp3 or.m4a) or the sampling rate was converted to 44,100 Hertz. **Fig K. Results of a Linear Discriminant Analysis trained on towhee song data.** Plot of LD1 results of a subset of towhee song bouts ($N_{test}=697$) using a Linear Discriminant Analysis trained on raw song feature data from a balanced training set of Spotted and Eastern towhee bouts ($N_{train}=1592$). The model revealed 86.8% prediction accuracy (balanced accuracy=86.9%; Cohen's κ=0.73). Points are jittered for visualization and color-coded by conversion status. 'Converted' denotes samples in which the original recording file was converted to a.wav file (from.mp3 or.m4a) or the sampling rate was converted to 44,100 Hertz. **Fig L. Species distributions of different audio file properties.** (**A**) We observe that wav files are biased toward the eastern U.S. (and thus the Eastern towhee) and mp3 files are biased toward the western U.S. (and thus the Spotted towhee). (**B**) This file-type bias corresponds to a difference in the proportion of submissions to Macaulay library in the east (which favors wav files) and Xeno-canto in the west (which allows users to download files in mp3 format). (**C**) The vast majority of files were either already sampled at 44,100 Hz (the standard for CD audio) or were sampled at 48,000 Hz (the standard for video).
(PDF)

## Author contributions

**Conceptualization:** Ximena León Du'Mottuchi, Nicole Creanza.

**Data curation:** Ximena León Du'Mottuchi.

**Formal analysis:** Ximena León Du'Mottuchi, Nicole Creanza.

**Funding acquisition:** Ximena León Du'Mottuchi, Nicole Creanza.

**Investigation:** Ximena León Du'Mottuchi, Nicole Creanza.

**Methodology:** Ximena León Du'Mottuchi, Nicole Creanza.

**Project administration:** Nicole Creanza.

**Resources:** Ximena León Du'Mottuchi, Nicole Creanza.

**Software:** Ximena León Du'Mottuchi, Nicole Creanza.

**Supervision:** Nicole Creanza.

**Validation:** Ximena León Du'Mottuchi, Nicole Creanza.

**Visualization:** Ximena León Du'Mottuchi, Nicole Creanza.

**Writing – original draft:** Ximena León Du'Mottuchi, Nicole Creanza.

**Writing – review & editing:** Ximena León Du'Mottuchi, Nicole Creanza.

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
