## [Decision Letter · Decision Letter 0]

Dear Dr. Creanza,

Thank you very much for submitting your manuscript "The role of learned song in the evolution and speciation of Eastern and Spotted towhees" for consideration at PLOS Computational Biology.

As with all papers reviewed by the journal, your manuscript was reviewed by members of the editorial board and by several independent reviewers. In light of the reviews (below this email), we would like to invite the resubmission of a significantly-revised version that takes into account the reviewers' comments.

I now got reports back from three reviewers who are experts in the field. As you will see in their comments, they found your study of interest and think that your manuscript is well written. However, they also raised some concerns that would have to be addressed before we can take a decision about your manuscript. Mainly, all three Reviewers commented on the analyses; the authors used a traditional machine learning method (random forest), which is a well established but low-power choice compared to more recently developed alternatives. Reviewer 1 also thinks that the use of three distinct approaches could be better justified. The Reviewers suggest several additional tests and alternative measures that could be computed. Secondly, all three Reviewer would also like to see the limitations of the study, in terms of the use of data from public websites, better acknowledged. This causes several potential issues, which could be tested or controlled for, and at least acknowledged in the discussion; the use of MP3 files, resampling sound clips with different sampling rates, public available sightings to estimate population demographics as well as potentially incorrect classification of hybrids, and no paired information of songs with morphology.

We cannot make any decision about publication until we have seen the revised manuscript and your response to the reviewers' comments. Your revised manuscript is also likely to be sent to reviewers for further evaluation.

Sincerely,

Elodie F Briefer

Guest Editor

PLOS Computational Biology

James O'Dwyer

Section Editor

PLOS Computational Biology

I now got reports back from three reviewers who are experts in the field. As you will see in their comments, they found your study of interest and think that your manuscript is well written. However, they also raised some concerns that would have to be addressed before we can take a decision about your manuscript. Mainly, all three Reviewers commented on the analyses; the authors used a traditional machine learning method (random forest), which is a well established but low-power choice compared to more recently developed alternatives. Reviewer 1 also thinks that the use of three distinct approaches could be better justified. The Reviewers suggest several additional tests and alternative measures that could be computed. Secondly, all three Reviewer would also like to see the limitations of the study, in terms of the use of data from public websites, better acknowledged. This causes several potential issues, which could be tested or controlled for, and at least acknowledged in the discussion; the use of MP3 files, resampling sound clips with different sampling rates, public available sightings to estimate population demographics as well as potentially incorrect classification of hybrids, and no paired information of songs with morphology.

Reviewer's Responses to Questions

**Comments to the Authors:**

Reviewer #1: the review was uploaded as an attachment

Reviewer #2: The authors investigate the calls of 2 sister species of songbirds with overlapping ranges. This overlap can lead to hybridization. In some cases, the characteristics of the specific calls are more extreme in these kinds of locations to avoid mating between the two species.

The authors use a range of different methods of analysis on the calls to find correlation between features and species/location. The also use a traditional machine learning method, random forest method, to show it can predict the species from its call.

The paper is clear and well written, and the figures are well made.

My major concern is that all the data (sound and sightings) for this study is obtained via public websites like xeno canto. This poses a few issues. First, resampling sound clips with different sampling rates can lead to less-than-optimal comparisons between the clips. The authors also use public available sightings to estimate population demographics. Sightings of the species of birds need to be objective and well planned to be used for population estimations. That has not happened here.

It need to be very clear from the discussion that there are limitations when using second hand data like it is the case here.

Fig 1 It would be informative to include frequency and time units on these spectrograms.

L 141 Using sounds from different data bases with different sampling frequencies can cause some problems with consistency. Because you save it in the same format and fs does not mean you get the same resolution. Aliasing can give you artifacts when saving at a different fs than the original. This can very much influence the correct extraction of max- and min syllable frequency for example. Then you need to show there is no correlation between fs to either species or location

L 166 It does not say how you arrive at those numbers and the supplementary figure does not give any new information. That figure needs to be improved with marking of the longitudes and better legend.

Basing your study solely on public sightings/recordings can give some problems. Do we know for sure the chance of reporting a sighting of the common species is the same as reporting of a rare breed/hybrid?

Supplementary figure 2 Why the discrepancy between the longitudes for the sightings and the vocalizations?

L 200 Why these numbers? Please explain this step better.

L 212 31 song bouts for hybrids is fairly low statistically.

L 221 That is the same number as before the log transformation except for the hybrid calls?

L 286 when using machine learning what made you use random tree approach instead of deep learning?

L 321 This section is just repeating part of the methods chapter.

L326 Using public sightings can lead to bias.

Table 1 Using colors for different categories like significant/non-significant makes reading such table much easier.

Fig 3 Are g and h really significantly different as indicated in the figure?

L 400 what is the 95 percent confidence interval for this?

L 427 Is this significantly different with such low number of individuals?

L 447 relatively narrow? What do you mean with that?

L 449 You really think just using eBird data gives you objective data to make such claim?

L 464 But 120 degrees does not really coincide with anything relating to the species, so your point is?

L 482 It is a bit unclear what we gain from the UMAP approach

496 I think the number of individuals might be in the low end to say anything about the trend in the hybrid zone?

Reviewer #3: This manuscript presents a straightforward and interesting analysis of variation in Eastern and Spotted towhee song throughout their overlapping ranges in North America. The analysis uses now standard methods like PCA, UMAP, and random forests, and the results are clear and well-justified by the authors.

My main criticism is that the authors restrict their machine learning analysis to random forests - a "tried and true" but also a fairly low-power choice compared to more recent alternatives like gradient-boosted machines and deep learning. If the machine learning were not the cornerstone of the paper then this would make sense, but papers in PLOS Computational Biology usually represent some sort of computational advancement and I do not see that in this paper. Both gradient-boosted machines and deep learning (via Torch) are now readily available in R from packages like gbm and cito, so the methods should be fairly easy to implement in this case.

I also have some more minor comments about the analysis below:

Conversion of MP3 to 44.1 kHz WAV is extremely lossy. Did you conduct an analysis to test whether the converted MP3 files vary in any spectral features with the WAV files? Or are there any other studies that show similar results?

Why were data log-transformed if Spearman rank correlations and Wilcoxon rank-sum tests were used? Isn't one of the big benefits of both of these methods that they can be used with non-normally distributed data, especially with a high sample size?

LDA is usually used instead of PCA when one wants to reduce dimensionality in a way that maximizes group differences. Based on the analysis code, it looks like the LDA was applied to the first two principal components rather than the raw data, which is atypical and unlikely to do much more than a simple linear model. I would strongly recommend trying out the LDA on all of the song features intead, as I think it would provide much better results!

On the UMAP analysis, as far as I know distances in UMAP embeddings are not as stable as those from PCA and other linear methods, and it's not recommended to analyze them statistically (e.g., see https://journals.plos.org/ploscompbiol/article?id=10.1371/journal.pcbi.1008228). This may not be an issue here, since the goal is simplify to use the distances in a classifier, but it might be worth mentioning.

The estimated rate of hybrids (0.4-1.6%) is extremely low. Do you think this might be an artifact of the use of eBird data? My intuition is that most eBird users are hobbyists, who are quite likely to incorrectly classify hybrids as either Eastern or Spotted towhees. Is there any way to estimate this? At the least, the potential for bias in the eBird data is something that should be acknowledged in the paper.

Figure 6 is great - I haven't seen this style of visual of model predictions before and it's very helpful!

**Have the authors made all data and (if applicable) computational code underlying the findings in their manuscript fully available?**

Reviewer #1: Yes

Reviewer #2: Yes

Reviewer #3: Yes

PLOS authors have the option to publish the peer review history of their article (what does this mean? ). If published, this will include your full peer review and any attached files.

**Do you want your identity to be public for this peer review?** For information about this choice, including consent withdrawal, please see our Privacy Policy .

Reviewer #1: No

Reviewer #2: No

Reviewer #3: No
---

## [Decision Letter · Decision Letter 1]

PCOMPBIOL-D-24-00199R1

The role of learned song in the evolution and speciation of Eastern and Spotted towhees

PLOS Computational Biology

Dear Dr. Creanza,

Thank you for submitting your manuscript to PLOS Computational Biology. After careful consideration, we feel that it has merit but does not fully meet PLOS Computational Biology's publication criteria as it currently stands. Therefore, we invite you to submit a revised version of the manuscript that addresses the points raised during the review process.

Please submit your revised manuscript within 30 days Apr 07 2025 11:59PM. If you will need more time than this to complete your revisions, please reply to this message or contact the journal office at ploscompbiol@plos.org. Please include the following items when submitting your revised manuscript:

We look forward to receiving your revised manuscript.

Kind regards,

Elodie F Briefer

Guest Editor

PLOS Computational Biology

Zhaolei Zhang

Section Editor

PLOS Computational Biology

**Additional Editor Comments :**

Based on my own reading of your revisions, I can see that you have made a lot of efforts in integrating all the previous comments of the three reviewers, particularly with the addition of two new machine learning algorithms. I sent back your revised manuscript to two of the former reviewers, and they have some comments left that would need to be addressed before we can take a decision about your manuscript. Reviewer 3 has only one minor comment left; namely, they would like you to extend the discussion on the effect of the use of mp3 files and necessary conversion. Reviewer 2 raises more comments, notably about the lack of explanations and justifications regarding commands used to run the new algorithms, as well as more details about how you control for the conversion effect you found.

**Reviewers' comments:**

Reviewer's Responses to Questions

Reviewer #2: General issues

The authors have done a lot of good work to improve the manuscript. Most importantly they do warn about the pitfalls of using a public data source without any real scientific quality control

The text can still be vague in certain locations, please be a bit more specific in your formulations.

Please do not just write what command you used in your software, explain the effect and why you did it. One of many examples is in the Deep Learning section where you state what command you used, not how you used it, what it does, or why you did it. It looks like you implemented cross entropy loss in your classification layer, but that is normally only done when you have an unbalanced data set. That is not the case with your data.

Please when you write your response make sure the line numbers are the real line numbers. It makes it hard for us reviewers to find the relevant text when the numbers are wrong.

In first round I was concerned about the conversion of sound from MP3 to Wav. Now you have shown it does play a role in your analysis and statistically significantly so. Can you then conclude anything from your tests when you cannot exclude that abiotic factor? As an example, have you investigated how much conversion means for PC1 and PC2? It does not seem like you have tried to isolate the impact of this factor on the various tests, only determined there is a significant effect.

Specific comments

Fig 1: You normally put more explanation in the figure itself. Having to have to read the figure text to get the axis labels and units is not good enough. This is just screenshots, please include appropriate labeling of axis.

L187 The conversion issue again. When converting from one format to another you can minimize issues by ensuring basic procedures are done, like ensuring a low pass filter of half the new sampling frequency is used. You then avoid aliasing. Please state what you have done to ensure best conversion.

L196 This formulation for choice of range is still vague like I commented in the first round.

L217 “we adjusted the signal-to-noise threshold, minimum syllable duration, and minimum silence duration to most accurately define the syllables in the song” How did you do that? By some parameter or more subjectively?

Fig 2 Please make the time visualization and text bigger. It is extremely hard to read what it says.

L291 That equation does not look right in this PDF

L361 This section there is a lot of “We then used command X” without any explanation of what that command does and why you did it.

L363 When I suggested Deep Learning I was thinking that you could do it either on the sound, or on the spectrograms, not the features you already extracted. That is the strength of DL networks: To find the relevant parameters by itself. This way you can look at the data from a completely different angle.

L364 You start stating that you used an artificial neural network with several hidden layers. That is a good example of what I mean with vague text. Write what network you used and how many layers there is.

L369 You claim to use a network with 64 layers as that is a multiple of the 16 features. To my knowledge there is no reason to use layers that is a multiple of the number of features you are investigating. Instead, you add layers if the data becomes more complicated.

L600 In the abstract you say the accuracy of Deep Learning is 89 percent, but here you say 90 percent.

Reviewer #3: The authors have done a fantastic job of addressing my original comments, adding two new machine learning analyses, and improving the strength of the other analyses. I only have one more minor comment below.

The effects of file conversion have higher p-values (so less significant than other factors) but they also have fairly similar t-values. This indicates that the amplitude of the effect of conversion is at least in the same order of magnitude as other factors, despite being "less significant". I do not think this is a big deal in terms of the interpretations of the results, but the authors should mention it. It may also be briefly worth discussing these effects, as they are relevant to other bioacousticians (I myself am considering mixing WAV and MP3 files in an analysis, and seeing the results of this paper would inform that decision).

**Have the authors made all data and (if applicable) computational code underlying the findings in their manuscript fully available?**

Reviewer #2: Yes

Reviewer #3: Yes

PLOS authors have the option to publish the peer review history of their article (what does this mean? ). If published, this will include your full peer review and any attached files.

**Do you want your identity to be public for this peer review?** For information about this choice, including consent withdrawal, please see our Privacy Policy .

Reviewer #2: No

Reviewer #3: No

**Figure resubmission:**
---

## [Editor Report · Decision Letter 2]

Dear Dr. Creanza,

We are pleased to inform you that your manuscript 'The role of learned song in the evolution and speciation of Eastern and Spotted towhees' has been provisionally accepted for publication in PLOS Computational Biology.

Best regards,

Elodie F Briefer

Guest Editor

PLOS Computational Biology

Zhaolei Zhang

Section Editor

PLOS Computational Biology

You provided a very thorough and comprehensive revision of your paper and addressed all the remaining comments of the reviewers adequately. I am thus pleased to accept your manuscript for publication.

---

## [Editor Report · Acceptance letter]

PCOMPBIOL-D-24-00199R2

The role of learned song in the evolution and speciation of Eastern and Spotted towhees

Dear Dr Creanza,

I am pleased to inform you that your manuscript has been formally accepted for publication in PLOS Computational Biology. Your manuscript is now with our production department and you will be notified of the publication date in due course.

With kind regards,

Anita Estes
